# CoLT: The conditional localization test for assessing the accuracy of neural posterior estimates

**Tianyu Chen**[*]**, Vansh Bansal**[*]
Department of Statstics and Data Sciences
UT Austin
{tianyuchen, vansh}@utexas.edu

**James G. Scott**
Department of Statistics and Data Sciences
McCombs School of Business
UT Austin
james.scott@mccombs.utexas.edu

## Abstract

We consider the problem of validating whether a neural posterior estimate $q(\theta \mid x)$ is an accurate approximation to the true, unknown true posterior $p(\theta \mid x)$. Existing methods for evaluating the quality of an NPE estimate are largely derived from classifier-based tests or divergence measures, but these suffer from several practical drawbacks. As an alternative, we introduce the *Conditional Localization Test* (CoLT), a principled method designed to detect discrepancies between $p(\theta \mid x)$ and $q(\theta \mid x)$ across the full range of conditioning inputs. Rather than relying on exhaustive comparisons or density estimation at every $x$, CoLT learns a localization function that adaptively selects points $\theta_l(x)$ where the neural posterior $q$ deviates most strongly from the true posterior $p$ for that $x$. This approach is particularly advantageous in typical simulation-based inference settings, where only a single draw $\theta \sim p(\theta \mid x)$ from the true posterior is observed for each conditioning input, but where the neural posterior $q(\theta \mid x)$ can be sampled an arbitrary number of times. Our theoretical results establish necessary and sufficient conditions for assessing distributional equality across all $x$, offering both rigorous guarantees and practical scalability. Empirically, we demonstrate that CoLT not only performs better than existing methods at comparing $p$ and $q$, but also pinpoints regions of significant divergence, providing actionable insights for model refinement. These properties position CoLT as a state-of-the-art solution for validating neural posterior estimates.

## 1 Introduction

This paper proposes a new method for determining whether two conditional distributions $p(\theta \mid x)$ and $q(\theta \mid x)$ are equal, or at least close, across all conditioning inputs. One of the most important applications of this idea arises in validating conditional generative models for neural posterior estimation, or NPE, which is a rapidly growing area of simulation-based inference. Here $\theta$ represents the parameter of a scientific model with prior $p(\theta)$, while $x \sim p(x \mid \theta)$ represents data assumed to have arisen from that model. In NPE, we simulate data pairs $(\theta, x)$ drawn from the joint distribution $(x, \theta) \sim p(x, \theta) \equiv p(\theta)p(x \mid \theta)$. A conditional generative model—such as a variational autoencoder [1], normalizing flow [2], diffusion model [3, 4, 5, 6], or flow-matching estimator [7]—is then trained on these $(x, \theta)$ pairs to approximate $p(\theta \mid x)$ with a learned distribution $q(\theta \mid x)$. The problem of neural posterior validation is to assess whether the learned $q$ is a good approximation to the true $p$.

This setting poses challenges not present in simpler problem of testing for equality of unconditional distributions, with no $x$. For one thing, we must verify that $q(\theta \mid x)$ approximates $p(\theta \mid x)$ not merely for a single given $x$, but consistently for all $x$, without having to explicitly consider all possible $x$

---

[*]equal contribution

39th Conference on Neural Information Processing Systems (NeurIPS 2025).

points. Moreover, most practical problems present a severe asymmetry in the available number of samples from $p$ and $q$. In NPE, for example, we observe just a single "real" sample $\theta \sim p(\theta \mid x)$ for each $x$, yet we can generate an arbitrary number of "synthetic" samples $\tilde{\theta} \sim q(\theta \mid x)$ by repeatedly querying our NPE model. Any successful method for assessing distributional equivalence of $p$ and $q$ must account for this imbalance.

**Existing methods.** Several methods have been proposed to assess the accuracy of a neural posterior estimate. But each has shortcomings. One popular method called Simulation-Based Calibration (SBC) [8] uses a simple rank-based statistic for each margin of $q(\theta \mid x)$, but this provides only a necessary (not sufficient) condition for posterior validity. Moreover, since rank statistics are computed separately for each margin, the statistical power of SBC suffers badly from multiple-testing issues in high-dimensional settings. TARP [9] provides a condition that is both necessary and sufficient for the neural posterior estimate to be valid. However, TARP's practical effectiveness depends heavily on the choice of a (non-trainable) probability distribution to generate "reference" points that are needed to perform the diagnostic, and the method can perform poorly under a suboptimal choice of this distribution. Finally, the classifer two-sample test, or C2ST [10], involves training a classifier to distinguish whether a given $\theta$ sample originates from the true posterior or the estimated one. It then uses the classifier output to construct an asymptotically normal test statistic under the null hypothesis that $p = q$. But as many others have observed, the C2ST hinges on the classifier's ability to effectively learn a global decision boundary over $\mathcal{X}$ and $\Theta$ simultaneously. In practice, the classifier may struggle to do so, due to insufficient training data, limited model capacity, or the inherent complexity of the task. Moreover, to perform well, the C2ST usually needs a class-balanced sample, which entails multiple draws of $\theta$ from the true posterior at a given $x$. This is often impractical, as in many settings we only have access to a single $\theta \sim p(\theta \mid x)$ at a given $x$.

**Our contributions.** Our paper addresses these shortcomings with a principled and efficient approach, called the *Conditional Localization Test* (CoLT), for detecting discrepancies between $p$ and $q$. CoLT is based on the principle of measure theoretic distinguishability: intuitively, if two conditional densities $p(\theta \mid x)$ and $q(\theta \mid x)$ are unequal, they must exhibit a nonzero difference in mass over some specific ball of positive radius. The basic idea of CoLT is to find that ball—that is, to train a *localization function* $\theta_l : \mathcal{X} \to \Theta$ that adaptively selects the point $\theta_l(x)$ where, for a given $x$, $p$ and $q$ are maximally different in the mass they assign to a neighborhood of $\theta_l(x)$. Intuitively, a neural network that learns a smooth mapping $\theta_l(x)$ should be well suited for this task: if two conditioning inputs $x$ and $\tilde{x}$ are close, we might reasonably expect that any differences between $p$ and $q$ would manifest similarly (i.e. in nearby regions of $\theta$ space) for both $x$ and $\tilde{x}$. This smoothness allows the network to generalize local differences across nearby regions in $x$ space, making the search for discrepancies both efficient and robust.

Of course, the principle of measure-theoretic distinguishability is well established, and so one might fairly ask: why has it not been widely exploited in machine learning as a tool for comparing conditional distributions? This is likely for two reasons, one geometric and one computational, both of which CoLT successfully addresses.

First, directly comparing mass over high-dimensional Euclidean balls can be ineffective for testing, as the Euclidean metric may not align with the geometry of how $p$ and $q$ are most readily distinguishable. To address this, we use a trainable embedding function $\phi$ that maps points from the parameter space $\Theta$ into a latent Euclidean space, where distances can better reflect the concentration of probability mass. We then assess mass equivalence over Euclidean balls in this latent space, i.e. over balls $B_\phi(\theta, R) = \{\theta' \in \Theta : \|\phi(\theta') - \phi(\theta)\|_2 \leq R\}$. We show how the necessary machinery from real analysis can be rigorously adapted to this setting, with modest requirements on $\phi$.

Second, even when assessing equivalence over non-Euclidean metric balls, naively training a localization function $\theta_l(x)$ would seem to require repeatedly sampling $\theta$ from both $p$ and $q$ at some $x$, comparing their local (Monte Carlo) integrals over all possible balls. This is intractable for all but the smallest problems. Luckily, we show that training $\theta_l$ can be done *far* more efficiently. The essential idea involves using a single observed draw from $p(\theta \mid x)$ to anchor our comparison of whether the conditional mass of $q(\theta \mid x)$ aligns with $p(\theta \mid x)$, in expectation over $x$. This single draw, combined with the localization function $\theta_l$, can be used to carefully construct a one-dimensional *ball probability rank* statistic that is uniformly distributed if and only if $p$ and $q$ agree on all local neighborhoods around $\theta_l(x)$. We rigorously construct this rank statistic, and we show how it leads to a practical

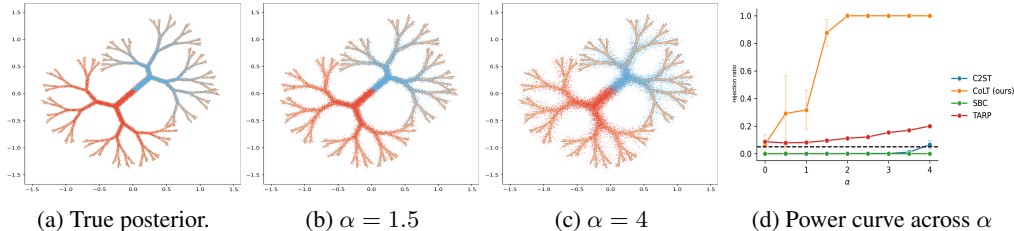

| (a) True posterior. | (b) $\alpha = 1.5$ | (c) $\alpha = 4$ | (d) Power curve across $\alpha$ |

Figure 1: Results on the toy tree-shaped example. As $\alpha$ increases (larger perturbation), the distribution becomes blurrier and deviates from the true manifold shown in Panel A. CoLT with a learned metric embedding maintains strong statistical power even for modest perturbations (Panel B), whereas the C2ST, SBC, and TARP all perform poorly even for much larger ones like $\alpha = 4$ (Panels C/D).

optimization algorithm for $\theta_l(x)$. Moreover, the rank statistic naturally induces a valid integral probability metric (IPM), offering a continuous measure of the distance between the two distributions. This is especially valuable in NPE settings: by moving beyond binary assessment, CoLT allows user of NPE methods to quantify improvements across training runs, benchmark multiple posterior approximators, or make targeted improvements to model architecture based on where specifically the neural posterior $q$ is performing poorly.

Finally, our empirical results demonstrate that CoLT consistently outperforms current state-of-the-art methods across a wide range of benchmark problems. The evidence shows that CoLT is able to consistently identify subtle discrepancies that classifier-based approaches routinely miss, providing strong empirical support for our theoretical analysis.

**A toy example.** To provide an initial demonstration of CoLT's effectiveness, we begin with a toy example. Panel A of Figure 1 shows $p(\theta \mid x)$ as living on a structured manifold, with branches A (bottom left) and B (top right) representing distinct regions of probability mass, as introduced in [11]. We sample a conditioning input as $x \sim \mathcal{N}(0, 1)$, with the true conditional distribution defined as:

$$p(\theta \mid x) \sim \begin{cases} \text{Branch A} & \text{if } x \geq 0, \\ \text{Branch B} & \text{if } x < 0. \end{cases}$$

Our goal here is to assess whether a method can reliably detect even small perturbations of $p$. This example, while simple, effectively targets a common failure mode of generative models: producing samples that lie near, but not exactly on, the true manifold of the posterior.

To benchmark CoLT's performance, we constructed "perturbed" posterior samples $\tilde{\theta} \sim q(\theta \mid x)$ by adding a small amount of isotropic Gaussian noise to "correct" samples: that is, $\tilde{\theta} = \theta + e$, where $\theta$ is a draw from $p(\theta \mid x)$ and each component of $e$ has standard deviation $0.01 \cdot \alpha$. We then varied $\alpha$, which controls the degree of mismatch between $p$ and $q$, and we tested the power of CoLT versus TARP, SBC, and the C2ST for each $\alpha$. The nominal Type-I error rate was set to 0.05 for all methods. To ensure a fair comparison, we trained the C2ST classifier and CoLT localization function with similar model capacities (number of layers and size of each layer); see Appendix C for details.

When $\alpha = 1.5$ (Panel B), the samples from $q$ fall very slightly off the correct manifold. CoLT can reliably detect this difference (power = 0.877), while C2ST failed entirely (power = 0.000). At a larger value of $\alpha = 4.0$ (Panel C), CoLT achieves perfect power (1.000), whereas C2ST only reaches power of 0.065. Panel D shows that, while performing a bit better than C2ST, neither TARP nor SBC are competitive with CoLT at any $\alpha$. These results highlight our method's performance advantage even in scenarios where the posterior lives on a structured manifold, and the discrepancy between $p$ and $q$ is reasonably small. We also emphasize that CoLT doesn't merely *detect* the difference; as our theory shows, it can also quantify the difference via an integral probability metric.

## 2 Theoretical Results

In this section, we present our main theoretical results; all proofs are given in the Appendix. Throughout, we denote the Lebesgue measure by $m(\cdot)$ and use $d\theta$ to represent Lebesgue integration. We also use the shorthand notation $q(\theta \mid x) = p(\theta \mid x)$, or simply $p = q$, to denote that $q(\theta \mid x) = p(\theta \mid x)$ for almost every $(\theta, x) \in \Theta \times \mathcal{X}$. Throughout, we assume that $p$ and $q$ are absolutely continuous with respect to Lebesgue measure for all $x$.

## 2.1 The conditional localization principle

CoLT relies on what we might call the *localization principle*: to check whether $p$ and $q$ are different, search for the point $\theta_l$, and the local neighborhood around $\theta_l$, where the mass discrepancy between $p$ and $q$ is as large as possible. If the largest such discrepancy is 0, the two distributions are equal.

Taken at face value, however, the localization principle seems deeply impractical. First, if we wish to conclude that $p(\theta \mid x) = q(\theta \mid x)$ for all $x$, it seems that we would need to apply the principle pointwise over a grid of $x$-values. Second, for each $x$, we would need to search for the point $\theta_l$ that maximizes the discrepancy in local mass between $p$ and $q$, if one exists for that $x$. Finally, we would need to draw many samples from both $p(\theta \mid x)$ and $q(\theta \mid x)$ to obtain reliable Monte Carlo integrals. The sheer number of evaluations needed—across many $x$-values, many candidate $\theta_l$-locations per $x$, and many Monte Carlo samples per $(\theta_l, x)$ pair—renders this naïve approach not just intractable, but *nestedly* intractable.

Luckily, we can do much better than the naïve approach. In fact, our subsequent results can be thought of as peeling back these layers of intractability one at a time.

We begin with a key definition. Specifically, we consider balls of the form $B_\phi(\theta, R) = \{\theta' \in \Theta : \|\phi(\theta') - \phi(\theta)\|_2 \le R\}$, where $\phi$ is an embedding function. By defining neighborhoods through $d_\phi$, we can shape our regions of comparison to better reflect meaningful differences in probability mass. The following imposes a mild, but useful, geometric regularity condition on the metric $\phi$.

**Definition 1** (Doubling Condition). *Let $\Theta$ be a set equipped with a map $\phi$ and let $m$ be a measure on $\Theta$. For each $\theta \in \Theta \subseteq \mathbb{R}^D$ and $R > 0$, define the $\phi$-ball*

$$B_\phi(\theta, R) = \{\theta' \in \Theta : \|\phi(\theta') - \phi(\theta)\|_2 \le R\}.$$

*We say that $\phi$ satisfies the doubling condition with respect to $m$ if there exists a constant $C > 0$ such that for all $\theta \in \Theta$ and all $R > 0$,*

$$m\big(B_\phi(\theta, 2R)\big) \le C\, m\big(B_\phi(\theta, R)\big). \tag{1}$$

Intuitively, this condition ensures that the metric balls $B_\phi$ do not distort the underlying geometry of $\mathbb{R}^D$ too severely, for instance by creating regions of infinite density or measure.

A straightforward sufficient condition for this is that the embedding $\phi$ be bi-Lipschitz of any order, which guarantees the above global doubling condition. However, a strict bi-Lipschitz map is not necessary. As a more flexible and practical alternative, we can define $\phi(\theta) = k_\xi(\theta, \cdot)$ as a deep-kernel embedding [12], which uses any Lipschitz encoder $\xi : \mathbb{R}^D \to \mathbb{R}^m$ to extract features. The corresponding kernel-based distance, given by $\|\phi(\theta') - \phi(\theta)\|_2 = \sqrt{k_\xi(\theta, \theta) + k_\xi(\theta', \theta') - k_\xi(\theta, \theta')}$, yeilds a *local* doubling condition, defined in Appendix A. This weaker, local condition is sufficient for our following localization result to hold. We provide a detailed proof and discussion for both the bi-Lipschitz and deep-kernel cases in Appendix A.

With this definition in place, we can state our first result about the equality of conditional distributions. This result replaces the stringent requirement of verifying an equality-of-mass condition *for each* $x$ with a weaker condition that involves *averaging* over $x$. We formalize this idea in terms of a localization function $\theta_l(x) : \mathcal{X} \to \Theta$, which identifies the most informative localization point based on $x$. Intuitively, $\theta_l(x)$ serves as a witness to any potential discrepancy between $p(\theta \mid x)$ and $q(\theta \mid x)$.

**Theorem 1** (Conditional localization). *Let $p(\theta \mid x)$ and $q(\theta \mid x)$ be defined as before, and define the difference function $d_x(\theta) = p(\theta \mid x) - q(\theta \mid x)$. Let $d_\phi : \Theta \times \Theta \to \mathbb{R}^+$ be the distance function, in induced by the embedding map $\phi$, satisfying the doubling condition with respect to Lebesgue measure. Let $B_\phi(\theta_l(x), R)$ denote the $\phi$-ball of radius $R$ centered at $\theta_l(x)$. Assume further that $p(x) > 0$ is a density on $\mathcal{X}$ which is strictly positive almost everywhere.*

*If, for every measurable function $\theta_l : \mathcal{X} \to \Theta$ and every $R > 0$, we have*

$$\int_{\mathcal{X}} p(x) \left[ \int_{B_\phi(\theta_l(x), R)} d_x(\theta)\, \mathrm{d}\theta \right] \mathrm{d}x = 0,$$

*then $d_x(\theta) = 0$ for almost every $(x, \theta)$ in $\mathcal{X} \times \Theta$.*

A full proof is provided in Appendix B. The sketch is as follows: The theorem's assumption—that the *average* discrepancy over all $x$ is zero—is challenging because positive and negative discrepancies

could cancel. However, a crucial feature of the theorem is that the center of the metric ball, $\theta_l(x)$, is allowed to *depend on* $x$ via a localization map. The proof uses a *measurable selection* argument to construct this adversarial localization function $\theta_l(x)$ that, for each $x$, intentionally centers the ball $B_\phi$ in the region of *maximum* discrepancy. Applying the theorem's hypothesis to this "worst-case" selector forces this maximum discrepancy to be zero for almost every $x$, which implies the discrepancy is zero for *all* balls. From this, the Lebesgue Differentiation Theorem [13]—which applies due to the doubling condition—allows us to conclude that if the average difference over all shrinking balls is zero, the pointwise difference $p(\theta|x) - q(\theta|x)$ must itself be zero almost everywhere. Moreover, $p = q$ implies that the supremum of

$$\int_{\mathcal{X}} p(x) \left[ \int_{B_\phi(\theta_l(x),R)} \big(p(\theta \mid x) - q(\theta \mid x)\big)\, d\theta \right] dx$$

over all measurable choices of $\theta_l(\cdot)$ and all $R > 0$, must be 0. This gives us a natural target for optimization over the choice of the localization function $\theta_l(x)$.

## 2.2   The ball probability rank statistic: a practical condition for mass equivalence

Theorem 1 eliminates the need for an exhaustive search over $x$. But its direct application still appears to require many draws from both $p(\theta \mid x)$ and $q(\theta \mid x)$ to verify the equality of mass over metric balls. Testing this condition via Monte Carlo would typically involve repeatedly sampling $\theta$ from both distributions at the same $x$ and comparing their local integrals. This remains computationally demanding even in principle. Moreover, in the typical setup where this methodology might be applied, the situation is asymmetric: $p(x)$ and $p(\theta \mid x)$ correspond to a real unknown distribution that generated the training data, meaning that for any observed $x$, we often have access to only a single corresponding draw from $p(\theta \mid x)$. By contrast, $q(\theta \mid x)$ represents a (conditional) generative model that we can query arbitrarily many times for a given $x$. A practical formulation must leverage this structure by treating the single "true" $\theta$ draw as an anchor and evaluating whether the conditional mass of $q$ aligns with $p$ in expectation over $x$.

Our next result establishes precisely this adaptation, ensuring that the comparison suggested by Theorem 1 can be done feasibly. The basic idea is as follows: we can draw a random sample $(\theta^*, x) \sim p(\theta, x)$, compute the localization point $\theta_l(x)$, and let the radius be implicitly determined as $R(\theta^*) = d_\phi(\theta_l(x), \theta^*)$. As the number of samples gets large, this turns out to be equivalent to checking all radii in Theorem 1. We now formalize this equivalence below, temporarily dropping the dependence on the conditioning input $x$ to lighten the notation.

**Theorem 2.** *Let $p$ and $q$ be defined as above. Fix a reference point $\theta_l \in \Theta$, and define the metric ball*

$$B_r = \{\theta \in \Theta : d_\phi(\theta_l, \theta) \le r\}.$$

*For any $\theta^* \in \Theta$, define the* ball probability rank *under $q$ as*

$$U_q(\theta^*) = P_{\theta \sim q}\big(d_\phi(\theta_l, \theta) \le d_\phi(\theta_l, \theta^*)\big).$$

*Then, the condition that $p$ and $q$ assign the same probability to all balls centered at $\theta_l$, i.e.,*

$$p(B_r) = q(B_r) \quad \text{for all radii } 0 \le r \le \sup_{\theta' \in \Theta} d(\theta', \theta_l),$$

*is equivalent to the statement that, when $\theta^* \sim p$, the random variable $U_q(\theta^*)$ is uniformly distributed on $[0, 1]$. That is, checking whether, for all choices of $\theta_l$, $U_q(\theta^*) \sim Unif(0, 1)$ under $\theta^* \sim p$ is both necessary and sufficient for $p = q$.*

Intuitively, if $p$ and $q$ differ, then there must exist some point $\theta_l$ and some radius $R$ for which the two distributions assign different mass to the ball $B(\theta_l, R)$. This mismatch causes the distribution of $U_q(\theta^*)$ to deviate from uniformity when $\theta^* \sim p$. Conversely, if $U_q(\theta^*) \sim \text{Unif}(0, 1)$ under $\theta^* \sim p$ for every choice of $\theta_l$, then $p$ and $q$ must agree on the mass of all such balls, and hence be identical. Thus taken together, Theorems 1 and 2 collapse a daunting, high-dimensional equality-of-mass requirement into a one-dimensional uniformity condition that can serve as the basis for a tractable optimization problem.

## 2.3 From local-mass uniformity to an IPM

Theorem 2 shows comparing the ball–probability rank statistic $U_{q,x}$ to a uniform distribution gives us a test for whether $p = q$. The next result shows that, once we optimize over every allowable localization map $\theta_l$, every embedding $\phi$, and every ball radius, the same uniformity test yields an integral probability metric (IPM) that we call the *averaged conditionally localized distance* (ACLD). Concretely, let

$$\mathcal{B} = \Big\{ \mathbf{1}_{B_\phi(\theta_l(x), R)}(\theta) \ : \ \theta_l : \mathcal{X} \to \Theta, \ \phi : \Theta \to \mathbb{R}^m, \ R > 0 \Big\},$$

the class of indicator functions of metric balls whose centers depend on $x$. The corresponding IPM is

$$\mathrm{ACLD}(p, q) = \sup_{f \in \mathcal{B}} \Big| \mathbb{E}_{x \sim p(x)} \Big[ \mathbb{E}_{\theta \sim p(\theta|x)} f(\theta) - \mathbb{E}_{\theta \sim q(\theta|x)} f(\theta) \Big] \Big|.$$

Our next theorem connects this distance to the ball probability rank statistic from Theorem 2.

**Theorem 3** (Ball–probability IPM). *Let $p(\theta \mid x)$ and $q(\theta \mid x)$ be absolutely continuous conditional densities on a common parameter space $\Theta$ for $x \in \mathcal{X}$, and suppose $p(x) > 0$ a.e. on $\mathcal{X}$. For x-dependent) center $\theta_l(x) \in \Theta$ and for the metric $d_\phi(\theta, \theta') = \|\phi(\theta) - \phi(\theta')\|_2$ induced by an embedding $\phi : \Theta \to \mathbb{R}^m$ satisfying the doubling condition, define*

$$U_{q,x}(\theta^*) = \mathbb{P}_{\theta \sim q(\theta|x)} \Big\{ d_\phi\big(\theta_l(x), \theta\big) \leq d_\phi\big(\theta_l(x), \theta^*\big) \Big\}.$$

*Now let $\widetilde{d}(p, q)$ be the worst-case Kolmogorov distance of $U_{q,x}(\theta^*)$, averaged over $x$, from the Uniform distribution:*

$$\widetilde{d}(p, q) = \sup_{\theta_l, \phi, \alpha \in [0,1]} \Big| \mathbb{E}_{x \sim p(x)} \Big[ \mathbb{P}_{\theta^* \sim p(\theta|x)} \big\{ U_{q,x}(\theta^*) \leq \alpha \big\} - \alpha \Big] \Big|.$$

*Then*

$$\widetilde{d}(p, q) = \mathrm{ACLD}(p, q).$$

The theorem establishes that the largest possible deviation from uniformity that one can provoke in $U_{q,x}$, by freely choosing the localization function, embedding, and ball radius, is numerically identical to an IPM built from indicator balls. Hence training the localization network $\theta_l(x)$ to *maximise* the distance between $U_{q,x}$ and $\mathrm{U}(0, 1)$ is equivalent to computing $\mathrm{ACLD}(p, q)$. If the optimizer fails to increase this distance beyond sampling noise, we have empirical evidence that $q(\theta \mid x)$ has passed the full mass-equivalence test implied by Theorem 1. Conversely, if $U_{q,x}$ is *not* uniformly distributed, then its empirical KS distance to $U(0, 1)$ gives us both a $p$-value based on the classical KS test, *and* estimates a distance between $p$ and $q$.

## 3 The CoLT Algorithm

The key insight from Theorem 2 is that searching for an embedding $\phi$ and localization function $\theta_l(x)$ that *maximally distort* the ball probability rank statistic $U_{q,x}$ away from uniformity is equivalent to detecting regions where $q$ fails to match $p$. This forms the basis of our optimization procedure. We represent both the metric embedding $\phi$ and the localization function as neural networks, $\theta_l(x; \psi)$ with learnable parameters $\psi$. Our strategy is roughly as follows:

- **Generate a rank statistic:** Draw a minibatch of "anchor" points $(\theta_i^*, x_i)_{i=1}^B$ from $p(\theta, x)$, the true joint distribution. By construction, each $(\theta_i^* \mid x_i)$ has conditional distribution $p(\theta \mid x_i)$. For each anchor point $i$, sample $M$ synthetic draws $\{\tilde{\theta}_{ij}\}_{j=1}^M$ from $q(\theta \mid x_i)$, and compute the empirical ball probability rank statistic:

$$\hat{U}_i(\psi, \phi) = \frac{1}{M} \sum_{j=1}^M \mathbf{1} \Big[ d_\phi\big(\theta_l(x_i; \psi), \tilde{\theta}_{ij}\big) \leq d_\phi\big(\theta_l(x_i; \psi), \theta_i^*\big) \Big].$$

- **Measure non-uniformity:** As a loss, we use a negative divergence from a uniform distribution, $L(\psi, \phi) = -D(\hat{U}_i(\psi, \phi), \mathrm{Uniform})$. We discuss the choice of divergence below.

- **Optimize:** Gradient descent is applied to the loss function. If $p = q$, optimization will stall, as no choice of $(\phi, \theta_l(x; \psi))$ will yield substantial deviation from uniformity. Otherwise, the optimizer finds a localization map that exposes the failure of $q$.

This approach is detailed in Algorithm 1 (training phase) and Algorithm 2 (testing phase). We first apply Algorithm 1 to train the embedding network $\phi$ and localization network $\theta_l$, aiming to maximize the discrepancy between the empirical $\hat{U}_i$ values and the uniform distribution. Then with the trained networks and a test set of $\{(\theta_i, x_i)\}$, we compute a test statistic and corresponding $p$-value using the one-sample Kolmogorov-Smirnov (KS) test in Algorithm 2.

---

**Algorithm 1** Conditional Localization Test (CoLT): Training Phase

---

1: **procedure** CoLT($\{(\theta_i, x_i)\}_{i=1}^N \sim p(\theta \mid x)p(x)$, sampling distribution $q(\theta \mid x)$)
2:     Generate $K$ samples $\{\theta_{ij}\} \sim q(\theta \mid x_i)$ for $i \in [N], j \in [K]$
3:     Define $d_\phi(\theta, \theta') = \|\phi(\theta) - \phi(\theta')\|_2$
4:     Initialize $\phi, \theta_l(x, \psi)$ as neural networks
5:     **while** not converged **do**
6:         **for** $i = 1, \ldots, N$ **do**
7:             $\theta_l \leftarrow \theta_l(x_i; \psi)$
8:             $U_i = \frac{1}{K} \sum_{j=1}^K \mathbf{1}\{d_\phi(\theta_{ij}, \theta_l) < d_\phi(\theta_i, \theta_l)\}$
9:         **end for**
10:         $L(\psi, \phi) = -D(U_i, \text{Uniform})$                 //Maximize divergence
11:         Perform gradient update on $\psi, \phi$
12:     **end while**
13:     **Return** $\theta_l, \phi$
14: **end procedure**

---

**Algorithm 2** Conditional Localization Test (CoLT): Testing Phase

---

1: **procedure** CoLT($\{(\theta_i, x_i)\}_{i=1}^N \sim p(\theta \mid x)p(x)$, sampling distribution $q(\theta \mid x)$, $\theta_l, \phi$)
2:     Generate $K$ samples $\{\theta_{ij}\} \sim q(\theta \mid x_i)$ for $i \in [N], j \in [K]$
3:     Define $d_\phi(\theta, \theta') = \|\phi(\theta) - \phi(\theta')\|_2$
4:     **for** $i = 1, \ldots, N$ **do**
5:         $\theta_l \leftarrow \theta_l(x_i)$
6:         $U_i = \frac{1}{K} \sum_{j=1}^K \mathbf{1}\{d_\phi(\theta_{ij}, \theta_l) < d_\phi(\theta_i, \theta_l)\}$
7:     **end for**
8:     $t, p \leftarrow \text{KS test}(\{U_1, \ldots, U_N\}, \text{Uniform})$         //test statistic & $p$-value
9:     **Return** $t, p$
10: **end procedure**

---

We make three remarks about this algorithm. First, because $U_i$ involves an indicator function, gradients cannot propagate directly; we therefore use the Straight-Through Estimator (STE) trick [14] to enable gradient-based optimization. Second, we represent the distance embedding network $\phi$ as a neural network due to its flexibility and capacity to approximate a wide range of transformations. Moreover, neural networks are typically Lipschitz-continuous under mild conditions [15], which ensures that the doubling condition (Definition 1) is satisfied; see Appendix A. Alternatively, a fixed, non-trainable form of $\phi$ can be specified, and our theoretical guarantees will still hold, but power may be reduced. For example, setting $\phi$ as the identity reduces $d(\cdot, \cdot)$ to the $\ell_2$ distance.

Third, in Algorithm 1, various divergence measures can be used to quantify the discrepancy between the empirical distribution of rank statistics $U_i$ and the uniform distribution. While the Kolmogorov–Smirnov (KS) distance is a natural choice motivated directly by our theory, it is not ideal for gradient-based optimization, which would need to propagate gradients through sorting and max operations. To address this, we instead use Sinkhorn divergence [16], an entropy-regularized version of Wasserstein distance that retains geometric sensitivity while offering a smooth objective. Empirically, we find that Sinkhorn divergence leads to stable optimization and good performance. We emphasize that Sinkhorn divergence is used only during the training phase to learn the localization and embedding maps. At test time, we use the KS statistic, as suggested by our theory, to compute $p$-values based on the empirical rank distribution.

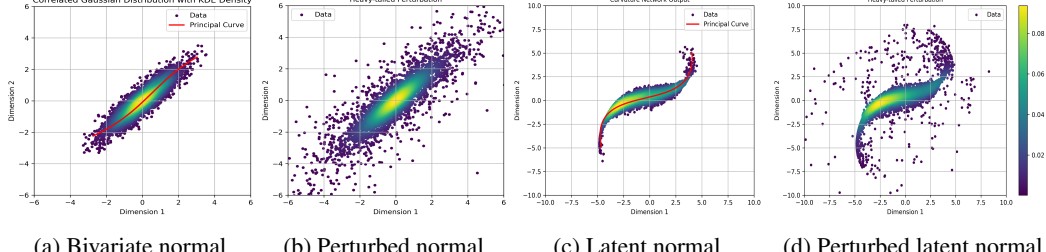

| (a) Bivariate normal | (b) Perturbed normal | (c) Latent normal | (d) Perturbed latent normal |

Figure 2: (A) Bivariate Gaussian with correlation 0.9. (B) Example perturbation of (A) to yield heavier tails. (C) Latent Gaussian with transformation. (D) Example perturbation of (C) with heavier tails in the latent space. In our benchmarks, CoLT and similar methods are tasked with distinguishing the ground-truth distributions (A and C) from perturbed variations (B and D, respectively). These are both large perturbations (large $\alpha$) and should be easy to detect; smaller $\alpha$ yields subtler perturbations. Details of the perturbation schemes are provided in Appendix C.2.

## 4 Experiments

**Benchmark tasks.** To evaluate CoLT against established NPT methods, we use a suite of benchmark tasks introduced by [17][2] (details in Appendix C.2). Each benchmark defines a reference posterior $p(\theta \mid x)$, then introduces a family of perturbed alternatives $q(\theta \mid x; \alpha)$, where the scalar parameter $\alpha \geq 0$ controls the severity of deviation. As $\alpha$ increases, so does the discrepancy between $p$ and $q$, allowing us to generate smooth performance curves that quantify the sensitivity of each NPT method.

We evaluate CoLT on two such benchmark families. The first is based on multivariate Gaussian posteriors with data-dependent mean and covariance. Specifically, we sample $x \sim \mathcal{N}(\mathbf{1}_m, I_m)$ and define

$$p(\theta \mid x) = \mathcal{N}(\mu_x, \Sigma_x), \quad \mu_x = W_1 x, \quad \Sigma_x = |W_2^\top x| \cdot \Sigma,$$

where $W_1 \in \mathbb{R}^{s \times m}$ and $W_2 \in \mathbb{R}^{s \times 1}$ are fixed matrices constructed from i.i.d. Gaussians, and $\Sigma$ is a Toeplitz matrix with entries $\Sigma_{ij} = \rho^{|i-j|}$, using $\rho = 0.9$. The alternative $q(\theta \mid x; \alpha)$ is then constructed by applying structured perturbations either to $\mu_x$ or $\Sigma_x$, as detailed in Appendix C.2. This setup allows us to simulate NPE errors such as mean shifts, covariance inflation, or distortions of multimodal structure. See Figure 2, Panels A-B.

The second family of benchmarks introduces geometric complexity by drawing latent Gaussian samples according to the same recipe as above, and then applying a nonlinear transformation, $\theta := f(\tilde{\theta}) \equiv A h(B\tilde{\theta})$, where $\tilde{\theta} \sim \mathcal{N}(\mu_x, \Sigma_x)$, $h$ is a coordinate-wise sine nonlinearity, and $A \in \mathcal{R}^{d \times d}$, $B \in \mathcal{R}^{d \times s}$ are fixed matrices. This creates a posterior distribution $p(\theta \mid x)$ concentrated on a smooth, curved manifold of intrinsic dimension $s$ in $\mathbb{R}^d$. To generate $q$, perturbations are applied in the latent Gaussian space (i.e. before transformation). See Figure 2, Panels C-D.

**Baselines and settings.** For our method, we evaluate two variants: **CoLT Full,** where both the embedding network $\phi$ and the localization network $\theta_l$ are jointly optimized; and **CoLT ID,** where $\phi$ is the identity and only the localization network $\theta_l$ is trained. We assess both Type I error (at $\alpha = 0$) and statistical power (for $\alpha > 0$) across all methods. Both versions of CoLT are compared against three established approaches: C2ST [10], SBC [8], and TARP [9]. To enable fair and meaningful comparisons, we adapt each baseline to produce a $p$-value, as follows. For C2ST, we sample one $\theta$ from $q(\theta \mid x)$ for each $x$ to create balanced training and test datasets, using the asymptotically normal test statistic described in [10]. For SBC, we conduct the KS test between the rank statistics and the uniform distribution for each dimension independently, followed by Bonferroni correction to control for multiple testing. For TARP, we select random reference points and the TARP test statistic to perform a KS test against the uniform distribution.

In both benchmark families, we vary the input, parameter, and latent dimensions $(m, s, d)$ and report power as a function of $\alpha$. We sample 100 pairs $\{(\theta_i, x_i)\}$ from the true joint distribution $p(\theta \mid x)p(x)$, along with 500 samples from $q(\theta \mid x)$ for each corresponding $x$ during training. After training, we evaluate a method's power by sampling 200 additional batches with the same sampling budget. For

---

[2] https://github.com/TianyuCodings/NPTBench

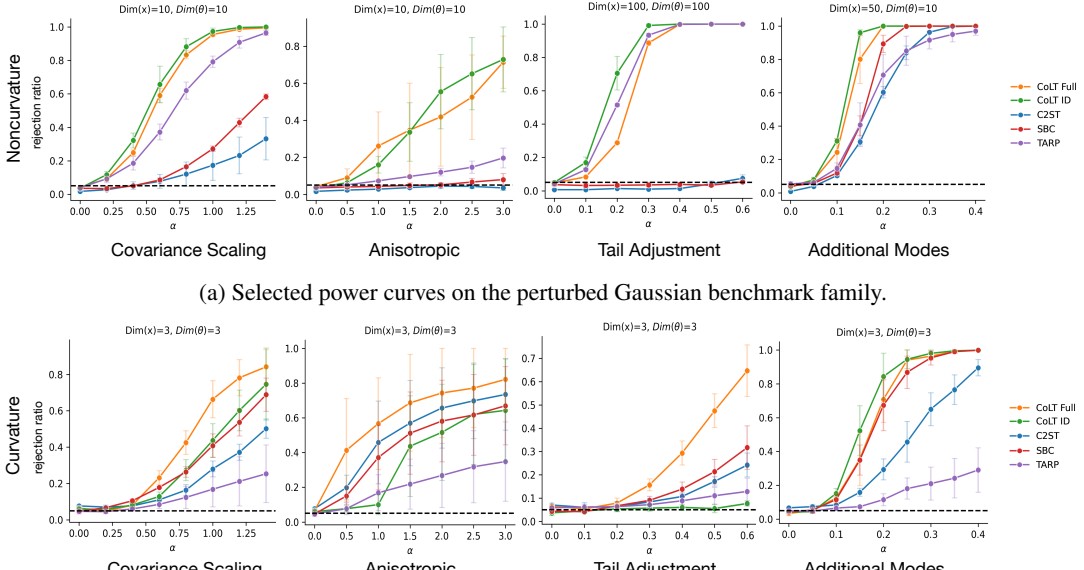

(a) Selected power curves on the perturbed Gaussian benchmark family.

(b) Selected power curves on the manifold benchmark family.

Figure 3: Statistical power curves (high is better) for four perturbation types under both benchmark families: (a) Gaussian posterior with data-dependent mean and covariance, and (b) its nonlinear transformation onto a curved manifold. Each panel refers to a specific perturbation type, with the horizontal axis ($\alpha$) describing the severity of the perturbation. Selected settings for $(s, d, m)$ are shown here, with results on a wider variety of settings shown in the Appendix C.2.

all methods, we set a nominal Type-I error rate of 5%. We repeat experiments with three random seeds and report averages. Further implementation details and design choices are in the Appendix C.

**Results.** Figure 3 summarizes the performance of the various testing methods across both benchmark families and four specific perturbation types: covariance scaling, anisotropic covariance distortion, heavy-tailed deviations via $t$-distributions, and the introduction of additional modes. In the simpler Gaussian benchmark (top row), both variants of CoLT (Full and ID) match or exceed the performance of C2ST while consistently outperforming SBC and TARP. CoLT ID—which measures mass over fixed Euclidean balls—performs well in cases like covariance scaling and additional modes, where the geometry of the discrepancy aligns well with the ambient space. C2ST also performs reasonably in these non-curved settings, particularly for tail adjustment and additional modes.

In contrast, the manifold benchmark (bottom row) reveals a clear advantage for CoLT Full, which learns a flexible embedding function to localize discrepancies. As with the toy example in Figure 1, this learned geometry appears essential in detecting errors, especially under tail adjustments and anisotropic distortions. CoLT ID, which lacks this geometric adaptability, performs notably worse than CoLT Full in these settings, although it still generally meets or exceeds the performance of other methods. These results highlight an important inductive bias: while fixed Euclidean balls suffice for flat posteriors, learned embeddings are crucial for detecting structured mismatch on curved or low-dimensional manifolds. Taken together, the results confirm that CoLT is competitive across a range of settings and is especially effective when for posteriors with complex geometry.

Additional experiments appear in Appendix C.3, which demonstrates our method's application to diffusion-based generative posteriors, and in Appendix D, which includes expanded results across more perturbation types and dimensional configurations. We provide the code at https://github.com/TianyuCodings/CoLT.

**Discussion and limitations.** Our theoretical and empirical results establish CoLT as a principled and practical approach for detecting local discrepancies between conditional distributions, with state-of-the-art performance compared to existing methods. But CoLT does have limitations. The method relies on learning both a localization function $\theta_l(x)$ and an embedding $\phi(\theta)$, introducing inductive

bias through architectural and optimization choices. If either component is underparameterized or poorly trained, CoLT may fail to detect real discrepancies. Its sensitivity also depends on the quality of the rank statistic, which can degrade with limited samples. And while CoLT yields a continuous IPM-style metric, interpreting this scalar, especially in high dimensions, can be challenging, as the underlying IPM function class is non-standard and implicitly defined by the learned components.

The benchmarking framework also has its limitations. Although designed to reflect realistic failure modes in NPE, the benchmarks are inherently synthetic and simplified. Perturbations are applied in controlled, parametric ways that may not capture the full complexity of real-world approximation errors. Moreover, the true posterior is always known, enabling rigorous evaluation but diverging from practical settings where ground truth is inaccessible. Despite these caveats, the suite provides a clear, extensible testbed, probing a number of common failure modes of NPE methods.

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

# A    Notes on $\phi$

**Bi-Lipschitz condition**    : The key requirement on $\phi$ is that it satisfies the doubling condition with respect to Lebesgue measure. One sufficient condition for this to hold is that $\phi$ be a bi-Lipschitz function, where there exist constants $C_1, C_2$ such that, for all $\theta_1, \theta_2$,

$$C_1 \|\theta_1 - \theta_2\|_2 \leq \|\phi(\theta_1) - \phi(\theta_2)\|_2 \leq C_2 \|\theta_1 - \theta_2\|_2$$

If this condition holds, then the doubling condition holds with doubling constant $C = \left(\frac{C_2}{C_1}\right)^D > 0$. To see this, observe that for such a $\phi$, the metric balls satisfy

$$B_{\text{Eucl}}(\theta_l, C_1 R) \subseteq B(\theta_l, R) \subseteq B_{\text{Eucl}}(\theta_l, C_2 R).$$

Then we have

$$m\big(B_{\text{Eucl}}(\theta_l, C_1 R)\big) = C_1^D \, m\big(B_{\text{Eucl}}(\theta_l, R)\big) \quad \text{and} \quad m\big(B_{\text{Eucl}}(\theta_l, C_2 R)\big) = C_2^D \, m\big(B_{\text{Eucl}}(\theta_l, R)\big).$$

Hence, from

$$B_{\text{Eucl}}(\theta, C_1 R) \subseteq B_\phi(\theta, R) \subseteq B_{\text{Eucl}}(\theta, C_2 R),$$

it follows that

$$C_1^D \, m\big(B_{\text{Eucl}}(\theta, R)\big) \leq m\big(B_\phi(\theta, R)\big) \leq C_2^D \, m\big(B_{\text{Eucl}}(\theta, R)\big),$$

which shows that $m\big(B_\phi(\theta, R)\big)$ scales like $R^D$ up to a constant factor.

**Deep kernel-based distances**    : We also propose using the following the deep kernel-based distance [12]:

$$d_\phi(\theta, \theta') = \|\phi(\theta) - \phi(\theta')\|^2 = \sqrt{k_\xi(\theta, \theta) + k_\xi(\theta', \theta') - 2k_\xi(\theta, \theta')} \tag{2}$$

where the infinite dimensional embedding $\phi(\theta) = k_\xi(\theta, \cdot) \in \mathcal{H}_k$ is defined through the following deep kernel:

$$k_\xi(\theta, \theta') = [(1 - \epsilon) \, k_1(\xi(\theta), \xi(\theta')) + \epsilon] \, k_2(\theta, \theta')$$

for $\epsilon \in (0, 1)$, RBF Gaussian kernels $k_1(\cdot, \cdot)$ and $k_2(\cdot, \cdot)$ given by $k_i(x, y) = \exp\left(\frac{\|x-y\|^2}{2\sigma_i^2}\right)$, and *any* Lipschitz embedding function $\xi : \mathbb{R}^D \to \mathbb{R}^m$. The kernel $k_\xi(\cdot, \cdot)$ is known to be a characteristic kernel [12], and hence defines a valid distance metric. The following lemma shows that the kernel-based distance defined in (2) satisfies a *local-doubling condition*, i.e., for any fixed radius $M > 0$, there exists a constant $C_M > 0$ such that for all $\theta \in \Theta$ and all radii $r \in (0, M]$, we have $\frac{m(B_\phi(\theta, 2r))}{m(B_\phi(\theta, r))} \leq C_M$.

**Lemma 1.** *Let $\xi : \mathbb{R}^D \to \mathbb{R}^m$ be a Lipschitz continuous function with constant $L_2$. The distance metric $d_\phi$ defined in (2) satisfies the following:*

1. *(Global Upper Bound) There exists a constant $C_1 > 0$ such that for all $\theta, \theta' \in \mathbb{R}^D$,*

$$d_\phi(\theta, \theta') \leq C_2 \|\theta - \theta'\|.$$

2. *(Local Lower Bound) For any $M > 0$, there exists a constant $C_M > 0$ such that for all $\theta, \theta' \in \mathbb{R}^D$ with $\|\theta - \theta'\| \leq M$,*

$$d_\phi(\theta, \theta') \geq C_{1,M} \|\theta - \theta'\|.$$

*Hence, $d_\phi$ is locally doubling as defined above.*

We defer the proof to Appendix B.5. The result of above lemma is critical as satisfying the local-doubling condition makes $(\Theta, d_\phi, m)$ a Vitali measure space [13, Theorem 3.4.3], which suffices for the Lebesgue Differentiation Theorem to hold and makes our Theorem 1 applicable.

# B Proofs

## B.1 A preliminary lemma

To prove Theorem 1 we first need the following lemma, which adapts standard measure-theoretic results to the case of a non-Euclidean metric based on an embedding function that satisfies the doubling condition in Definition 1.

**Lemma 2.** *Let $p(\theta \mid x)$ and $q(\theta \mid x)$ be defined as above, and let $\phi$ be an embedding function that induces a metric $d_\phi$ on $\Theta$, defined as*

$$d_\phi(\theta_1, \theta_2) = \|\phi(\theta_1) - \phi(\theta_2)\|_2.$$

*Further, assume that $\phi$ satisfies the doubling condition (1) with respect to Lebesgue measure.*

*Suppose that for almost every $\theta_l \in \Theta$, we have*

$$\int_{B(\theta_l, R)} p(\theta \mid x) \, d\theta = \int_{B(\theta_l, R)} q(\theta \mid x) \, d\theta$$

*for all metric balls $B(\theta_l, R)$, defined as*

$$B(\theta_l, R) = \{\theta \in \Theta : d_\phi(\theta, \theta_l) \leq R\}.$$

*Then $p(\theta \mid x) = q(\theta \mid x)$ almost everywhere ($\theta$).*

*Proof.* Define the *difference function*

$$d_x(\theta) = p(\theta \mid x) - q(\theta \mid x).$$

The goal is to show that $d_x(\theta) = 0$ almost everywhere in $\Theta$ using the given integral condition.

Because $\phi$ is assumed to satisfy the doubling condition with respect to Lebesgue measure, we have for some $C > 0$ that

$$m(B_\phi(\theta, 2R)) \leq Cm(B_\phi(\theta, R))$$

Now since $d_x(\theta)$ is locally integrable (as it is a difference of probability densities), we apply the Lebesgue Differentiation Theorem for doubling measures [18], which implies:

$$\lim_{R \to 0} \frac{1}{m(B_\phi(\theta, R))} \int_{B_\phi(\theta, R)} d_x(\theta') \, d\theta' = d_x(\theta) \quad \text{for a.e. } \theta.$$

However, by assumption, we know that for all $\theta$ and all sufficiently small $R$,

$$\int_{B_\phi(\theta, R)} d_x(\theta') \, d\theta' = 0.$$

Since $m(B(\theta, R)) > 0$, dividing by the Lebesgue measure of the ball and taking the limit yields:

$$d_x(\theta) = 0 \quad \text{for almost every } \theta.$$

Since $d_x(\theta) = 0$ a.e., it follows that $p(\theta \mid x) = q(\theta \mid x)$ almost everywhere in $\Theta$. $\qquad\square$

## B.2 Proof of Theorem 1

Let $B(\theta, R)$ be a $d_\phi$-ball with radius $R$. Assume the doubling property: $\exists\, C \geq 1$ such that

$$m\big(B(\theta, 2R)\big) \leq C\, m\big(B(\theta, R)\big) \quad \forall\, \theta \in \Theta, \ R > 0,$$

and that ball boundaries have $m$-measure zero.

Let $\mathcal{X}$ be an open subset of $\mathbb{R}^k$ with Borel measure $\mu$. Suppose

- $p : \mathcal{X} \to (0, \infty)$ is a continuous probability density;
- $(x, \theta) \mapsto p(\theta \mid x)$ and $(x, \theta) \mapsto q(\theta \mid x)$ are continuous in $x$ and belong to $L^1_{\text{loc}}(\Theta)$ for every $x$;

- for every measurable function $\theta_l : \mathcal{X} \to \Theta$ and every $R > 0$,

$$\int_{\mathcal{X}} p(x) \left[ \int_{B(\theta_l(x),R)} \big( p(\theta \mid x) - q(\theta \mid x) \big) d\theta \right] dx \; = \; 0.$$

Then we show that $p(\theta \mid x) = q(\theta \mid x)$ for $\mu \otimes m$-a.e. $(x, \theta)$.

*Proof.* The proof proceeds in two main steps. First, we fix $R > 0$ and use a measurable selection argument to show that the inner integral is zero for almost every $x$. Second, we apply the Lebesgue Differentiation Theorem to show that the integrand itself must be zero almost everywhere.

**Step 1: Show ball integrals are zero for a.e. $x$.** Let $d_x(\theta) = p(\theta \mid x) - q(\theta \mid x)$. For a fixed $R > 0$, define the ball integral

$$F_R(x, \theta) := \int_{B(\theta,R)} d_x(\theta') \, d\theta'.$$

The function $F_R(x, \theta)$ is a Carathéodory function: that is, it is measurable in $x$ for each fixed $\theta$ (by Fubini's theorem, as $d_x$ is continuous in $x$) and continuous in $\theta$ for each fixed $x$ (by the Dominated Convergence Theorem, as $d_x \in L^1_{\mathrm{loc}}$ and $m(\partial B(\theta, R)) = 0$).

Define the pointwise suprema:

$$S_R^+(x) := \sup_{\theta \in \Theta} F_R(x, \theta),$$
$$S_R^-(x) := \sup_{\theta \in \Theta} \big( -F_R(x, \theta) \big),$$
$$G_R(x) := \sup_{\theta \in \Theta} |F_R(x, \theta)| = \max\{S_R^+(x), S_R^-(x)\}.$$

Since $\Theta \subset \mathbb{R}^D$ is separable, let $D = \{\vartheta_k\}_{k \geq 1}$ be a fixed countable dense subset. Because $\theta \mapsto F_R(x, \theta)$ is continuous and $D$ is dense, the suprema over $\Theta$ are equal to the suprema over $D$. Specifically, let $\widetilde{S}_R^+(x) := \sup_{k \geq 1} F_R(x, \vartheta_k)$ and $\widetilde{S}_R^-(x) := \sup_{k \geq 1}(-F_R(x, \vartheta_k))$. These are measurable functions, since they are countable suprema of measurable functions, and we have $S_R^+(x) = \widetilde{S}_R^+(x)$ and $S_R^-(x) = \widetilde{S}_R^-(x)$ for all $x$.

Now, we construct measurable $\varepsilon$-maximizing selectors. Fix $n \in \mathbb{N}$. Define:

$$k_n^+(x) := \min\Big\{ k \geq 1 : \; F_R\big(x, \vartheta_k\big) \geq \widetilde{S}_R^+(x) - \tfrac{1}{n} \Big\},$$
$$k_n^-(x) := \min\Big\{ k \geq 1 : \; -F_R\big(x, \vartheta_k\big) \geq \widetilde{S}_R^-(x) - \tfrac{1}{n} \Big\}.$$

These minima are well-defined and finite. By the definition of the supremum, for any $\varepsilon = 1/n > 0$, the set of indices $k$ satisfying the condition is guaranteed to be non-empty. By the well-ordering principle, a non-empty subset of $\mathbb{N}$ has a minimum. The maps $x \mapsto k_n^\pm(x)$ are measurable, as the sets $\{x : k_n^+(x) = k\}$ are formed by measurable comparisons. Thus, the selectors $\theta_{R,n}^+(x) := \vartheta_{k_n^+(x)}$ and $\theta_{R,n}^-(x) := \vartheta_{k_n^-(x)}$ are well defined and measurable.

By construction, these selectors satisfy:

$$F_R\big(x, \theta_{R,n}^+(x)\big) \geq \widetilde{S}_R^+(x) - \tfrac{1}{n} \quad \text{and} \quad -F_R\big(x, \theta_{R,n}^-(x)\big) \geq \widetilde{S}_R^-(x) - \tfrac{1}{n} \quad \text{for all } x.$$

Applying the theorem's hypothesis with $\theta_l = \theta_{R,n}^+$ gives:

$$0 = \int_{\mathcal{X}} p(x) F_R\big(x, \theta_{R,n}^+(x)\big) \, dx \geq \int_{\mathcal{X}} p(x) \left( \widetilde{S}_R^+(x) - \tfrac{1}{n} \right) dx \implies \int_{\mathcal{X}} p(x) \widetilde{S}_R^+(x) \, dx \leq \frac{1}{n} \int_{\mathcal{X}} p(x) \, dx.$$

Similarly, applying the hypothesis with $\theta_l = \theta_{R,n}^-$ gives:

$$0 = \int_{\mathcal{X}} p(x) F_R\big(x, \theta_{R,n}^-(x)\big) \, dx \leq \int_{\mathcal{X}} p(x) \left( -\widetilde{S}_R^-(x) + \tfrac{1}{n} \right) dx \implies \int_{\mathcal{X}} p(x) \widetilde{S}_R^-(x) \, dx \leq \frac{1}{n} \int_{\mathcal{X}} p(x) \, dx.$$

Since $G_R(x) \leq S_R^+(x) + S_R^-(x) = \widetilde{S}_R^+(x) + \widetilde{S}_R^-(x)$, we have

$$\int_{\mathcal{X}} p(x) G_R(x) \, dx \leq \int_{\mathcal{X}} p(x) \widetilde{S}_R^+(x) \, dx + \int_{\mathcal{X}} p(x) \widetilde{S}_R^-(x) \, dx \leq \frac{2}{n} \int_{\mathcal{X}} p(x) \, dx = \frac{2}{n}.$$

This must hold for all $n$, and so we must have $\int_{\mathcal{X}} p(x) G_R(x) \, dx = 0$. Since $p(x) > 0$ and $G_R(x) \geq 0$, we must therefore have $G_R(x) = 0$ for $p$-a.e. $x$. This means that for $p$-a.e. $x$, we have $F_R(x, \theta) = 0$ for all $\theta \in \Theta$.

**Step 2: Apply the Lebesgue Differentiation Theorem.** From Step 1, we know there is a set $X_0 \subset \mathcal{X}$ with $p(X_0) = 1$ such that for any $x \in X_0$, $\int_{B(\theta, R)} d_x(\theta') \, d\theta' = 0$ for all $R > 0$ and all $\theta \in \Theta$.

The Lebesgue Differentiation Theorem for doubling spaces states that for any function $f \in L^1_{\mathrm{loc}}(\Theta, m)$,

$$f(\theta) = \lim_{R \to 0} \frac{1}{m(B(\theta, R))} \int_{B(\theta, R)} f(\theta') \, d\theta' \quad \text{for } m\text{-a.e. } \theta.$$

For any fixed $x \in X_0$, the function $\theta \mapsto d_x(\theta)$ is in $L^1_{\mathrm{loc}}(\Theta)$ by hypothesis. Applying the theorem gives:

$$d_x(\theta) = \lim_{R \to 0} \frac{1}{m(B(\theta, R))} \int_{B(\theta, R)} d_x(\theta') \, d\theta' = \lim_{R \to 0} \frac{0}{m(B(\theta, R))} = 0$$

for $m$-a.e. $\theta \in \Theta$.

Since this holds for every $x \in X_0$ where $\mu(\mathcal{X} \setminus X_0) = 0$ (as $p$ is a density for $\mu$), we have $d_x(\theta) = 0$ for $\mu \otimes m$-a.e. $(x, \theta)$. This means $p(\theta \mid x) = q(\theta \mid x)$ for $\mu \otimes m$-a.e. $(x, \theta)$.

**Remarks.** We conclude with two remarks.

1. **Necessity of quantifying over all measurable selectors.** The assumption that the inner integral vanishes for *every measurable* selector $\theta_l : \mathcal{X} \to \Theta$ is crucial. If the condition held only for constant maps $\theta_l(x) \equiv \theta_0$, it would assert only that the $x$-average of the localized integrals vanishes:

$$\int p(x) \left[ \int_{B(\theta_0, R)} d_x(\theta) \, d\theta \right] dx = 0.$$

This would allow for cancellations across $x$ and would not imply that the inner integral vanishes pointwise in $x$. The logic of the proof requires the freedom to vary the center of the ball *adaptively* with $x$ to prevent these cancellations.

2. **Continuity in $x$ can be relaxed.** The theorem and proof remain valid if the maps $x \mapsto p(\theta \mid x)$ and $x \mapsto q(\theta \mid x)$ are merely *measurable* rather than continuous, provided that the resulting function $F_R(x, \theta) = \int_{B(\theta, R)} d_x(\theta') \, d\theta'$ is a Carathéodory function (measurable in $x$, continuous in $\theta$). This condition holds under weaker assumptions than continuity, such as joint measurability of $(x, \theta) \mapsto d_x(\theta)$. The continuity-in-$x$ assumption is a straightforward condition that guarantees this property, which is all that is needed for the measurable selection argument to succeed. Moreover, the measurability of the set $N = \{(x, \theta) : x \in X_0, \theta \in \Theta_x^c\}$, where $\Theta_x^c$ is the set of non-Lebesgue points for the function $d_x(\theta) = d(x, \theta)$, follows from the joint measurability of $d_x(\theta)$.

$\square$

## B.3 Proof of Theorem 2

We first need the following lemma.

**Lemma 3.** *Let $(\Theta, d_\phi)$ be a metric space, and let $\theta_l \in \Theta$ be fixed. Define the function*

$$R(\theta) = d_\phi(\theta_l, \theta), \quad \theta \in \Theta.$$

*Now let $q$ be a probability measure on $\Theta$. For any $\theta^* \in \Theta$, define the **ball probability rank** of $\theta^\star$ under $q$ as*

$$U_q(\theta^*) = P_{\theta \sim q}\big(R(\theta) \le R(\theta^*)\big) = P_{\theta \sim q}\big(d_\phi(\theta_l, \theta) \le d_\phi(\theta_l, \theta^*)\big).$$

*Then, if we also have that $\theta^* \sim q$, then the random variable $U_q(\theta^*)$ is distributed as Uniform$(0, 1)$, i.e.,*

$$P_{\theta^* \sim q}\big(U_q(\theta^*) \le u\big) = u, \quad \forall u \in [0, 1].$$

*Proof.* Define $F_R$ as the cumulative distribution function (CDF) of the random variable $R_q(\theta) = d(\theta_l, \theta)$, where $\theta \sim q$, i.e.,

$$F_R(r) = P_{\theta \sim q}\big(R(\theta) \le r\big).$$

By definition of $U(\theta^*)$, we have

$$U_q(\theta^*) = P_{\theta \sim q}\big(R(\theta) \le R(\theta^*)\big) = F_R(R(\theta^*)).$$

But by assumption, we have $\theta^\star \sim q$. Accordingly, $R(\theta^*)$ is itself a random variable drawn from the distribution whose CDF is $F_R$, it follows from the probability integral transform that for any localization point $\theta_l$,

$$P_{\theta^\star \sim q}\big(U(\theta^\star) \le u\big) = u. \tag{3}$$

Thus, $U_q(\theta^*) \sim$ Uniform$(0, 1)$, completing the proof.

$\square$

The key observation from Lemma 3 is that the probability mass assigned by $p$ to the ball of this radius, centered at $\theta_l$, follows a uniform distribution when $\theta^* \sim p$. Thus, if equation (3) (which states that $U(\theta^*) \sim$ Uniform$(0, 1)$) holds for all possible choices of $\theta_l$, then the conditional distributions $p(\theta \mid x)$ and $q(\theta \mid x)$ must be identical. Intuitively, this is because the process of drawing $\theta^*$ and measuring probability mass within its corresponding ball implicitly tests equality of mass across all possible radii in a structured way. If the distributions $p$ and $q$ were different, there would exist some localization point $\theta_l$ where the resulting uniformity condition fails, revealing a discrepancy in their induced probability measures.

With this lemma in place, we can now prove Theorem 2.

*Proof.* ($\Rightarrow$) Suppose that $p(B_r) = q(B_r)$ for all $r \ge 0$. Consider the cumulative distribution function (CDF) of the distance variable $R(\theta^*) = d(\theta_l, \theta^*)$, when $\theta^* \sim p$:

$$F_p(r) = P_{\theta^* \sim p}\big(R(\theta^*) \le r\big) = p(B_r).$$

Similarly, under $\theta^* \sim q$, the corresponding CDF is

$$F_q(r) = P_{\theta^* \sim q}\big(R(\theta^*) \le r\big) = q(B_r).$$

By assumption, these two CDFs are identical, i.e., $F_p(r) = F_q(r)$ for all $r$. Now, by the definition of $U_q(\theta^*)$,

$$U_q(\theta^*) = P_{\theta \sim q}\big(R(\theta) \le R(\theta^*)\big) = F_q(R(\theta^*)).$$

Since $F_q = F_p$, we obtain

$$U_q(\theta^*) = F_p(R(\theta^*)).$$

From Lemma 3, we know that $F_p(R(\theta^*)) \sim$ Uniform$(0, 1)$ when $\theta^* \sim p$, which implies that $U_q(\theta^*) \sim$ Uniform$(0, 1)$ under $\theta^* \sim p$. Thus, the distributions of $U_q(\theta^*)$ under $p$ and $q$ must be identical.

($\Leftarrow$) Now suppose that $U_q(\theta^*) \sim U_p(\theta^*)$. Then, for any $u \in [0, 1]$,

$$P_{\theta^* \sim p}\big(U_q(\theta^*) \le u\big) = P_{\theta^* \sim q}\big(U_q(\theta^*) \le u\big).$$

Rewriting in terms of the CDFs, this implies

$$P_{\theta^* \sim p}\big(F_q(R(\theta^*)) \le u\big) = P_{\theta^* \sim q}\big(F_q(R(\theta^*)) \le u\big).$$

By the probability integral transform, since $F_q(R(\theta^*)) \sim$ Uniform$(0, 1)$ under both $p$ and $q$, it follows that $F_q(R(\theta^*)) = F_p(R(\theta^*))$ in distribution. This means that $F_p = F_q$, implying

$$p(B_r) = q(B_r), \quad \forall r.$$

Thus, the probability assigned to each metric ball is identical under $p$ and $q$.

$\square$

## B.4   Proof of Theorem 3

From Lemma 3, we have that $U_q(\theta^* \mid x)$ follows a uniform distribution when $\theta^* \sim q(\theta^* \mid x)$, thus satisfying

$$\mathbb{P}_{U \sim \text{Unif}(0,1)} [U \leq \alpha] = \mathbb{P}_{\theta^* \sim q(\theta^* \mid x)} [U_q(\theta^* \mid x) \leq \alpha].$$

Next, define the radius $R_{\theta_l(x)}(\alpha)$ as follows

$$R_{\theta_l(x)}(\alpha) := \inf \left\{ r : \mathbb{P}_{\theta \sim q(\theta \mid x)} [\theta \in B_\phi(\theta_l(x), r)] = \alpha \right\}.$$

Since $q(\theta \mid x)$ is absolutely continuous with respect to the Lebesgue measure, the mapping $\alpha \mapsto R_{\theta_l(x)}(\alpha)$ is a bijection. Additionally, by definition of $U_q(\theta^* \mid x)$, it follows that

$$\{\theta^* : U_q(\theta^* \mid x) \leq \alpha\} = B_\phi(\theta_l(x), R_{\theta_l(x)}(\alpha)).$$

Combining these observations, we obtain:

$$\tilde{d}(p, q) = \sup_{\theta_l(\cdot), \phi, \alpha} \left| \mathbb{E}_{x \sim p(x)} \left[ \mathbb{P}_{\theta^* \sim p(\theta^* \mid x)} [U_q(\theta^* \mid x) \leq \alpha] - \mathbb{P}_{\theta^* \sim q(\theta^* \mid x)} [U_q(\theta^* \mid x) \leq \alpha] \right] \right|$$

$$= \sup_{\theta_l(\cdot), \phi, \alpha} \left| \mathbb{E}_{x \sim p(x)} \left[ \mathbb{P}_{\theta^* \sim p(\theta^* \mid x)} \left[ B_\phi(\theta_l(x), R_{\theta_l(x)}(\alpha)) \right] - \mathbb{P}_{\theta^* \sim q(\theta^* \mid x)} \left[ B_\phi(\theta_l(x), R_{\theta_l(x)}(\alpha)) \right] \right] \right|$$

$$= \sup_{\theta_l(\cdot), \phi, R} \left| \mathbb{E}_{x \sim p(x)} \left[ \mathbb{P}_{\theta^* \sim p(\theta^* \mid x)} [B_\phi(\theta_l(x), R)] - \mathbb{P}_{\theta^* \sim q(\theta^* \mid x)} [B_\phi(\theta_l(x), R)] \right] \right| = \text{ACLD}(p, q)$$

## B.5   Proof of Lemma 1

*Proof.* Let $u = \frac{\|\theta - \theta'\|^2}{2\sigma_2^2}$ and $v = \frac{\|\phi(\theta) - \phi(\theta')\|^2}{2\sigma_1^2}$. Since $\theta, \theta' \in \mathbb{R}^D$, $u \geq 0$ and $v \geq 0$. The squared distance is given by

$$d_\phi^2(\theta, \theta') = 2 \left( 1 - \left[ (1 - \epsilon)e^{-v} + \epsilon \right] e^{-u} \right).$$

**Proof of the Global Upper Bound**   We can rewrite the expression for the squared distance as:

$$d_\phi^2(\theta, \theta') = 2 - 2(1 - \epsilon)e^{-(u+v)} - 2\epsilon e^{-u} = 2(1 - e^{-(u+v)}) - 2\epsilon e^{-u}(1 - e^{-v}).$$

For any $z \geq 0$, the standard inequality $1 - e^{-z} \leq z$ holds. Furthermore, since $\epsilon \in (0, 1)$, $e^{-u} > 0$, and $1 - e^{-v} \geq 0$, the second term is non-positive. We can therefore bound the expression:

$$d_\phi^2(\theta, \theta') \leq 2(1 - e^{-(u+v)}) \leq 2(u + v).$$

Substituting the definitions of $u$ and $v$ yields:

$$d_\phi^2(\theta, \theta') \leq 2 \left( \frac{\|\theta - \theta'\|^2}{2\sigma_2^2} + \frac{\|\phi(\theta) - \phi(\theta')\|^2}{2\sigma_1^2} \right) = \frac{\|\theta - \theta'\|^2}{\sigma_2^2} + \frac{\|\phi(\theta) - \phi(\theta')\|^2}{\sigma_1^2}.$$

By the Lipschitz assumption on $\phi$, we have $\|\phi(\theta) - \phi(\theta')\| \leq L_2 \|\theta - \theta'\|$, which implies:

$$d_\phi^2(\theta, \theta') \leq \frac{1}{\sigma^2} \left( \|\theta - \theta'\|^2 + L_2^2 \|\theta - \theta'\|^2 \right) = \left( \frac{1}{\sigma_2^2} + \frac{L_2^2}{\sigma_1^2} \right) \|\theta - \theta'\|^2.$$

Taking the square root provides the global upper bound with the constant $C_2 = \sqrt{\frac{1}{\sigma_2^2} + \frac{L_2^2}{\sigma_1^2}}$.

**Proof of the Local Lower Bound**   We rewrite the term $1 - k_\xi(\theta, \theta')$:

$$1 - k_\xi(\theta, \theta') = 1 - \left[ (1 - \epsilon)e^{-v} + \epsilon \right] e^{-u} = (1 - e^{-u}) + (1 - \epsilon)e^{-u}(1 - e^{-v}).$$

Since $1 - \epsilon > 0$, $e^{-u} > 0$, and $1 - e^{-v} \geq 0$, the second term is non-negative. This allows us to bound the expression below:

$$1 - k_\xi(\theta, \theta') \geq 1 - e^{-u}.$$

This provides a lower bound on the squared distance:

$$d_\phi^2(\theta, \theta') = 2(1 - k_\xi(\theta, \theta')) \geq 2(1 - e^{-u}) = 2\left(1 - \exp\left(-\frac{\|\theta - \theta'\|^2}{2\sigma_2^2}\right)\right).$$

To establish a linear relationship between $d_\phi(\theta, \theta')$ and $\|\theta - \theta'\|$, we analyze the function $g(z) = \frac{1 - e^{-z}}{z}$ for $z > 0$. By L'Hôpital's rule, $\lim_{z \to 0^+} g(z) = 1$. The function $g(z)$ is continuous and strictly positive on any compact interval $[0, Z_{\max}]$. By the Extreme Value Theorem, it must attain a minimum value $c_{\min} > 0$ on this interval. Therefore, for all $z \in (0, Z_{\max}]$, the inequality $1 - e^{-z} \geq c_{\min} \cdot z$ holds.

Let us restrict our domain to a bounded set where $\|\theta - \theta'\| \leq M$ for some constant $M > 0$. This implies that $u = \frac{\|\theta - \theta'\|^2}{2\sigma^2} \leq \frac{M^2}{2\sigma^2} = Z_{\max}$. On this domain, we can apply the linear inequality derived above:

$$d_\phi^2(\theta, \theta') \geq 2(1 - e^{-u}) \geq 2c_{\min}u = 2c_{\min}\frac{\|\theta - \theta'\|^2}{2\sigma_2^2} = \frac{c_{\min}}{\sigma_2^2}\|\theta - \theta'\|^2.$$

Taking the square root gives the local lower bound $d_\phi(\theta, \theta') \geq C_{1,M}\|\theta - \theta'\|$ with the constant $C_{1,M} = \frac{\sqrt{c_{\min}}}{\sigma_2}$, which depends on $M$ through $c_{\min}$.

This yields the local-doubling condition with constant $C_M = \left(\frac{C_2}{C_{1,M}}\right)^D$, completing the proof. $\square$

## C  Experiments Details

### C.1  Toy Example

We construct a synthetic ground-truth data distribution in $\mathbb{R}^2$ by defining a Gaussian mixture model (GMM) whose components are procedurally placed according to a recursive branching process. We use the code from paper [11] whose generation process can be found in https://github.com/NVlabs/edm2/blob/main/toy_example.py. To maintain the completeness of our paper, we include the generation process here.

**Gaussian Mixture Representation**

The base distribution is modeled as a weighted sum of multivariate Gaussian components:

$$p(x) = \sum_{k=1}^{K} \phi_k \, \mathcal{N}(x \mid \mu_k, \Sigma_k + \sigma^2 I),$$

where:

- $\phi_k \in \mathbb{R}_+$ are normalized mixture weights,
- $\mu_k \in \mathbb{R}^2$ are the component means,
- $\Sigma_k \in \mathbb{R}^{2\times 2}$ are the component covariance matrices,
- $\sigma \in \mathbb{R}_+$ where we set $\sigma = 1\mathrm{e} - 2$ for $p(\theta \mid x)$ and $(1 + \alpha) \cdot \sigma$ for $q(\theta \mid x)$.

Each component is assigned a weight and covariance that decays with tree depth, producing finer-scale detail at deeper recursion levels.

**Recursive Tree-Structured Composition**

The mixture components are positioned according to a recursive tree-like structure:

- Two primary classes (A and B) are generated, each initialized at the same root origin and with distinct initial angles (e.g., $\pi/4$ and $5\pi/4$).
- At each recursion level (up to depth 7), a branch is extended in a given direction, and eight Gaussian components are placed uniformly along the branch.

Table 1: Six types of perturbations used to assess the sensitivity of CoLT.

| $p(\theta \mid x)$ | $q(\theta \mid x)$ | Explanation |
|---|---|---|
| $\mathcal{N}(\mu_x, \Sigma_x)$ | $\mathcal{N}((1+\alpha)\mu_x, \Sigma_x)$ | **Mean Shift:** Introduces a systematic bias by shifting the mean. |
| | $\mathcal{N}(\mu_x, (1+\alpha)\Sigma_x)$ | **Covariance Scaling:** Uniformly inflates the variance. |
| | $\mathcal{N}(\mu_x, \Sigma_x + \alpha\Delta)$ | **Anisotropic Covariance Perturbation:** Adds variability along the minimum-variance eigenvector of $\Sigma_x$: $\Delta = \mathbf{v}_{\min}\mathbf{v}_{\min}^\top$. |
| | $t_\nu(\mu_x, \Sigma_x)$ | **Tail Adjustment via $t$-Distribution:** Introduces heavier tails, with degrees of freedom $\nu = 1/(\alpha + \epsilon)$, approaching Gaussian as $\alpha \to 0$. |
| | $(1-\alpha)\mathcal{N}(\mu_x, \Sigma_x) + \alpha\,\mathcal{N}(-\mu_x, \Sigma_x)$ | **Additional Modes:** $q$ introduces spurious multimodality. |
| $(1-\alpha)\mathcal{N}(\mu_x, \Sigma_x) + \alpha\,\mathcal{N}(-\mu_x, \Sigma_x)$ | $\mathcal{N}(\mu_x, \Sigma_x)$ | **Mode Collapse:** $q$ loses multi-modal structure. |

- Each component's mean is computed by interpolating along the current direction vector, and the covariance is anisotropically scaled to align with the branch's orientation.
- Each branch spawns two child branches recursively, with angles perturbed stochastically to simulate natural variability.

**Component Covariance Structure**

The covariance of each Gaussian component is constructed as:

$$\Sigma = \left(\mathbf{d}\mathbf{d}^\top + (\mathbf{I} - \mathbf{d}\mathbf{d}^\top) \cdot \text{thick}^2\right) \cdot \text{diag}(\text{size})^2,$$

where $\mathbf{d}$ is the normalized direction of the branch, `thick` controls orthogonal spread, and `size` scales with recursion depth.

This construction ensures that components are elongated along the branch direction and narrow orthogonal to it, producing tree-like density patterns.

## C.2 Experiments with perturbed Gaussians

We give further details on the experiments in Section 4. For these benchmarks, we construct a ground-truth conditional distribution by first simulating latent Gaussian variables with $x$-dependent means and variances:

$$\tilde{\theta} \sim \mathcal{N}(\mu_x, \Sigma_x), \quad \mu_x = W_1 x, \quad \Sigma_x = |W_2^\top x| \cdot \Sigma,$$

where $\tilde{\theta} \in \mathbb{R}^s$, $W_1 \in \mathbb{R}^{s \times m}$, and $W_2 \in \mathbb{R}^{s \times 1}$ are fixed weight matrices with standard normal entries. The matrix $\Sigma \in \mathbb{R}^{s \times s}$ is a fixed Toeplitz correlation matrix, with entries $\Sigma_{ij} = \text{corr}^{|i-j|}$ and $\text{corr} = 0.9$ to simulate strong structured correlations.

In all cases we sample conditioning inputs $x \sim \mathcal{N}(\mathbf{1}_m, I_m)$.

## C.3 Diffusion Training and Sampling Procedure

**Diffusion Model**  To further evaluate our method in the context of generative posterior estimation, we construct a benchmark based on a diffusion model. We begin by sampling $x \sim \text{Unif}[1.5\pi, 4.5\pi]$ and defining $\theta = (x\cos(x), x\sin(x))$, which induces a highly nonlinear and non-Gaussian posterior structure. We then train a diffusion model to approximate this distribution. During evaluation, we generate samples via reverse diffusion and treat the output after 20 reverse steps as a notional ground-truth posterior (Figure 4a)—not because it is the true target distribution, but because it represents the best available approximation produced by the model. Outputs from earlier steps $(20 - \alpha)$ serve as degraded approximations to this endpoint. This yields a generative-model-based, monotonic perturbation scheme parameterized by $\alpha$ (Figures 4b and 4c).

Beyond simple rejection, our method provides fine-grained quantitative insight: we use the test statistic $t$ from Algorithm 2 to measure how close each approximate posterior (from fewer reverse steps) is to the reference posterior (20-step output). This allows us to quantify posterior degradation as a function of reverse diffusion progress. As shown in Figure 4d, the test statistic increases monotonically with $\alpha$, reflecting growing divergence from the true posterior.

As described in Section 4, we sample $x \sim \text{Unif}[1.5\pi, 4.5\pi]$ and define $\theta = (x\cos(x), x\sin(x))$. This produces a nontrivial two-dimensional manifold for posterior inference.

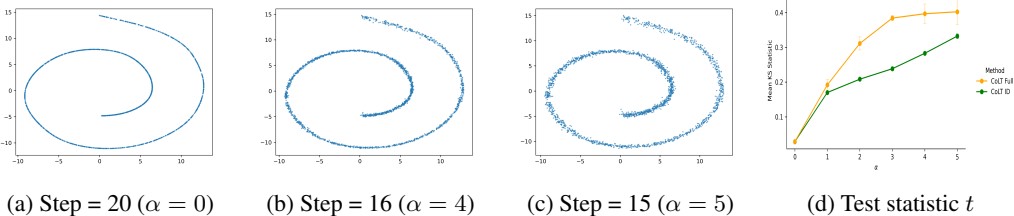

(a) Step = 20 ($\alpha = 0$)  (b) Step = 16 ($\alpha = 4$)  (c) Step = 15 ($\alpha = 5$)  (d) Test statistic $t$

Figure 4: Posterior estimation using a diffusion model. The output at step 20 is treated as the ground-truth posterior. Reducing the number of reverse steps results in increasingly degraded approximations. Our method reflects this degradation with a monotonically increasing test statistic, indicating sensitivity to model quality.

We generate 32,768 samples to train a diffusion model using 50,000 training epochs and a learning rate of $1 \times 10^{-4}$. The diffusion model is trained and sampled following the implementation provided in EDM [19].

For posterior approximation, we set the total number of reverse diffusion steps to 20, treating the output at step 20 as the ground-truth posterior. To construct approximate posteriors at varying levels of fidelity, we also record intermediate samples from reverse steps 15 through 19. These intermediate outputs serve as posterior estimates for evaluation against the step-20 reference.

## C.4   Sampling and Training Hyperparameters

In this section, we provide detailed configurations for the data generation process and training setup used throughout our experiments (see Section 4). We evaluate our method across various combinations of (dim($x$),dim($\theta$)): $(3, 3)$, $(10, 10)$, $(50, 10)$, and $(100, 100)$. We denote $N$ as the number of sample pairs $\{(\theta_i, x_i)\}$ drawn from the true joint distribution $p(\theta \mid x)p(x)$, and $K$ as the number of samples drawn from the estimated posterior $q(\theta \mid x)$ for each conditioning value $x$.

In the **CoLT Full** setting, we utilize a distance embedding network $\phi$ with input dimension equal to $\dim(\theta)$ and output dimension set to $\dim(\theta)$. Although alternative output dimensions for $\phi$ may potentially improve performance, we fix the output dimension to $\dim(\theta)$ to avoid additional hyperparameter tuning and ensure a fair comparison across settings.

All neural networks in our method (including $\phi$, $\theta_l$ and C2ST classifier) are implemented as 3-layer multilayer perceptrons (MLPs) with 256 hidden units per layer.

The table below summarizes the range of perturbation levels $\alpha$ tested for each experiment type, along with the sampling and training hyperparameters for both CoLT and C2ST.

| Perturbation | Alphas | $N$ | $K$ | #Eval | CoLT Epochs | CoLT LR | C2ST Epochs | C2ST LR |
|---|---|---|---|---|---|---|---|---|
| Mean Shift | (0.0, 0.05, 0.1, 0.15, 0.2, 0.25, 0.3) | 100 | 500 | 200 | 25 | 1e−5 | 1000 | 1e−5 |
| Covariance Scaling | (0.0, 0.2, 0.4, 0.6, 0.8, 1.0, 1.2, 1.4) | 100 | 500 | 200 | 1000 | 1e−5 | 1000 | 1e−5 |
| Anisotropic Perturbation | (0.0, 0.5, 1.0, 1.5, 2.0, 2.5, 3.0) | 100 | 500 | 200 | 1000 | 1e−5 | 1000 | 1e−5 |
| Kurtosis Adjustment via $t$-Distribution | (0.0, 0.1, 0.2, 0.3, 0.4, 0.5, 0.6) | 100 | 500 | 200 | 1000 | 1e−5 | 1000 | 1e−5 |
| Additional Modes | (0.0, 0.05, 0.1, 0.15, 0.2, 0.25, 0.3, 0.35, 0.4) | 100 | 500 | 200 | 1000 | 5e−5 | 1000 | 5e−5 |
| Mode Collapse | (0.0, 0.1, 0.2, 0.3, 0.4, 0.5, 0.6) | 100 | 500 | 200 | 1000 | 1e−5 | 1000 | 1e−5 |
| Blind Prior | — | 100 | 500 | 200 | 1000 | 1e−3 | 1000 | 1e−5 |
| Tree (Toy Example) | (0.0, 0.5, 1.0, 1.5, 2.0, 2.5, 3.0, 3.5, 4.0) | 1000 | 100 | 200 | 5000 | 1e−5 | 5000 | 1e−5 |
| Diffusion | (0, 1, 2, 3, 4, 5) | 1000 | 200 | 200 | 1000 | 1e−5 | 1000 | 1e−5 |

Table 2: Experimental configurations for each type of posterior perturbation. Columns specify the perturbation type, tested $\alpha$ values, sample counts, evaluation batch size, and training hyperparameters for CoLT and C2ST methods.

## C.5   Curvature Transformation and Calculation

In Section 4, we introduce the concept of curvature in the parameter space. Specifically, we construct a transformation network to increase the curvature of $\theta$. The network consists of a fully connected layer ('torch.nn.Linear') with input dimension equal to the dimension of $\theta$, a hidden layer of

size 128, followed by an element-wise sine activation, and another linear layer mapping from 128 to the original dimension of $\theta$. The weights of the linear layers are initialized using PyTorch's default random initialization.

To compute the curvature, we apply the principal curve algorithm from Hastie and Stuetzle [20]. The resulting principal curves are shown in Figures 2a and 2c.

We observe that before applying the curvature-inducing transformation, the principal curve closely resembles a straight line, with a total absolute curvature of approximately 62, as expected for a highly correlated Gaussian distribution. After applying the transformation, the resulting parameter space exhibits significantly increased curvature, with a total absolute curvature of around 400.

This transformation provides an effective mechanism for inducing curvature in $\theta$ space, allowing us to study the performance of methods under non-Euclidean geometries.

# D    Additional Experimental Results

## D.1    Tree Task

We present additional visualizations for the toy tree-structured posterior under various levels of perturbation $\alpha$. As $\alpha$ increases, the sampled points become increasingly dispersed and less concentrated around the underlying structure. The shaded region indicates the true posterior manifold corresponding to $\alpha = 0$.

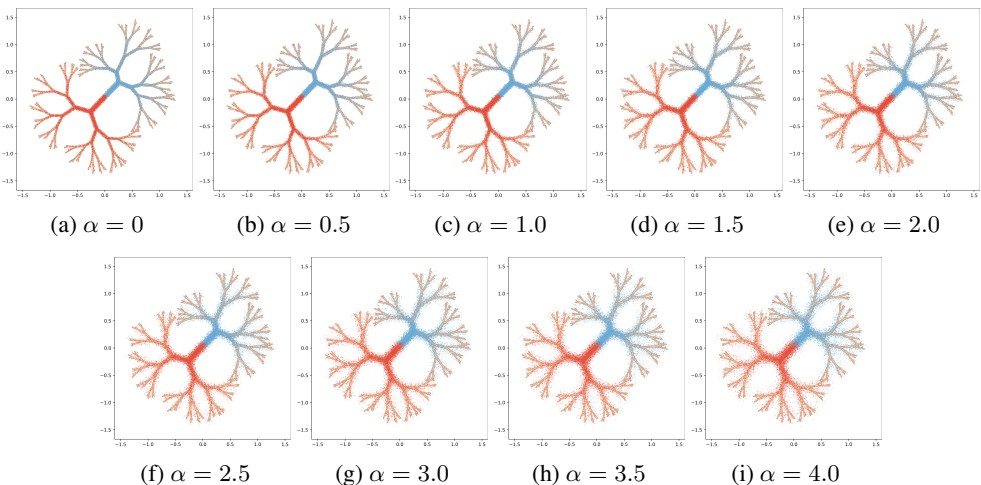

(a) $\alpha = 0$     (b) $\alpha = 0.5$     (c) $\alpha = 1.0$     (d) $\alpha = 1.5$     (e) $\alpha = 2.0$

(f) $\alpha = 2.5$     (g) $\alpha = 3.0$     (h) $\alpha = 3.5$     (i) $\alpha = 4.0$

Figure 5: Tree samples across varying $\alpha$ values.

The statistical power for different methods, including **CoLT Full** and **CoLT ID**, is shown in Figure 6.

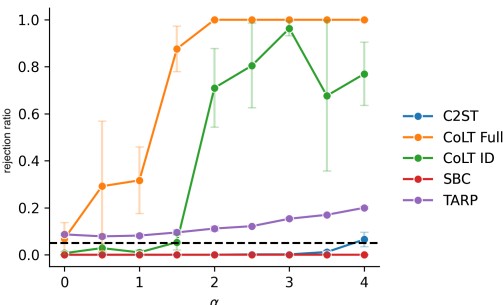

Figure 6: Statistical power for tree-structured tasks across all evaluated methods, including CoLT variants.

## D.2 Perturbation

In this section, we present additional results across varying dimensions of $(x, \theta)$ for different tasks, evaluated under multiple perturbation magnitudes $\alpha$ in the *non-curvature* setting. The results demonstrate the robustness of CoLT and baseline methods under a wide range of perturbations.

Specifically, we show:

- **Mean shifts**: Figure 8
- **Covariance scaling**: Figure 9
- **Anisotropic covariance perturbation**: Figure 10
- **Kurtosis adjustment via $t$-distribution**: Figure 11
- **Additional modes**: Figure 12
- **Mode collapse**: Figure 13

## D.3 Blind Prior

In addition to the above perturbation strategies, we also include a **Blind Prior** setting, where the estimated posterior ignores the input $x$ entirely: $q(\theta \mid x) = p(\theta)$, i.e., the posterior estimate is simply the prior distribution of $\theta$. This scenario serves as an important pathological case, as it has been shown to cause both SBC and TARP (with random reference points) to fail—these methods are unable to detect the distributional discrepancy between $q(\theta \mid x)$ and the true posterior $p(\theta \mid x)$. By contrast, we demonstrate that our proposed method remains sensitive and effective even in this setting.

**Blind Prior**   In Table 3, we present results under the Blind Prior setting, where the estimated posterior ignores the conditioning input and is set as $q(\theta \mid x) = p(\theta)$. This case is particularly challenging, as both TARP and SBC fail to detect the resulting distributional discrepancy.

In contrast, our proposed methods—CoLT ID and CoLT Full—successfully detect this violation across all dimensional settings. Notably, while C2ST is effective in low dimensions, its power deteriorates significantly as the dimensionality increases. Our methods maintain high power even in high-dimensional regimes, demonstrating their robustness and effectiveness in detecting subtle posterior mismatches.

| Method | $x = 3, \theta = 3$ | $x = 10, \theta = 10$ | $x = 50, \theta = 10$ | $x = 100, \theta = 100$ |
|---|---|---|---|---|
| CoLT ID | **1.000** $\pm$ 0.000 | **1.000** $\pm$ 0.000 | **1.000** $\pm$ 0.000 | **1.000** $\pm$ 0.000 |
| CoLT Full | 0.975 $\pm$ 0.014 | 0.778 $\pm$ 0.222 | 0.693 $\pm$ 0.307 | 0.452 $\pm$ 0.260 |
| C2ST | **1.000** $\pm$ 0.000 | **1.000** $\pm$ 0.000 | 0.847 $\pm$ 0.038 | 0.122 $\pm$ 0.007 |
| SBC | 0.052 $\pm$ 0.004 | 0.028 $\pm$ 0.007 | 0.048 $\pm$ 0.007 | 0.040 $\pm$ 0.015 |
| TARP | 0.053 $\pm$ 0.004 | 0.047 $\pm$ 0.012 | 0.035 $\pm$ 0.009 | 0.068 $\pm$ 0.004 |

Table 3: Statistical power (mean $\pm$ stderr) under the Blind Prior setting for each method, evaluated across increasing dimensions. Only CoLT variants consistently maintain high power as dimensionality increases.

In addition to reporting the statistical power of each method in Table 3, we provide their corresponding Type I error rates in Table 4. Since the $p$-value threshold is set to 0.05, all methods successfully control the Type I error within the expected range, indicating that none falsely reject the null hypothesis under the correctly specified posterior.

| Method | $x = 3, \theta = 3$ | $x = 10, \theta = 10$ | $x = 50, \theta = 10$ | $x = 100, \theta = 100$ |
|---|---|---|---|---|
| C2ST | 0.0767 $\pm$ 0.0044 | 0.0433 $\pm$ 0.0067 | 0.0200 $\pm$ 0.0050 | 0.0233 $\pm$ 0.0073 |
| CoLT Full | 0.0400 $\pm$ 0.0076 | 0.0400 $\pm$ 0.0029 | 0.0517 $\pm$ 0.0109 | 0.0467 $\pm$ 0.0093 |
| CoLT ID | 0.0567 $\pm$ 0.0093 | 0.0550 $\pm$ 0.0076 | 0.0467 $\pm$ 0.0044 | 0.0433 $\pm$ 0.0017 |
| SBC | 0.0400 $\pm$ 0.0104 | 0.0350 $\pm$ 0.0000 | 0.0350 $\pm$ 0.0000 | 0.0350 $\pm$ 0.0087 |
| TARP | 0.0367 $\pm$ 0.0093 | 0.0383 $\pm$ 0.0017 | 0.0517 $\pm$ 0.0109 | 0.0433 $\pm$ 0.0017 |

Table 4: Type I error (mean $\pm$ stderr) under the Blind Prior setting for each method, evaluated across increasing dimensions.

### D.4 Diffusion Sampling Results

In this section, we provide additional visualizations related to the diffusion-based posterior approximation. Figure 7 illustrates the underlying data manifold used to train the diffusion model, as well as the sampling results from reverse steps 15 through 20. These samples allow us to visualize the quality of intermediate outputs as the reverse process progresses.

We also include a power and Type I error curve that quantifies how the performance of our method changes with respect to the number of reverse steps. As expected, the statistical power increases as the number of reverse steps approaches 20, while Type I error remains well-controlled.

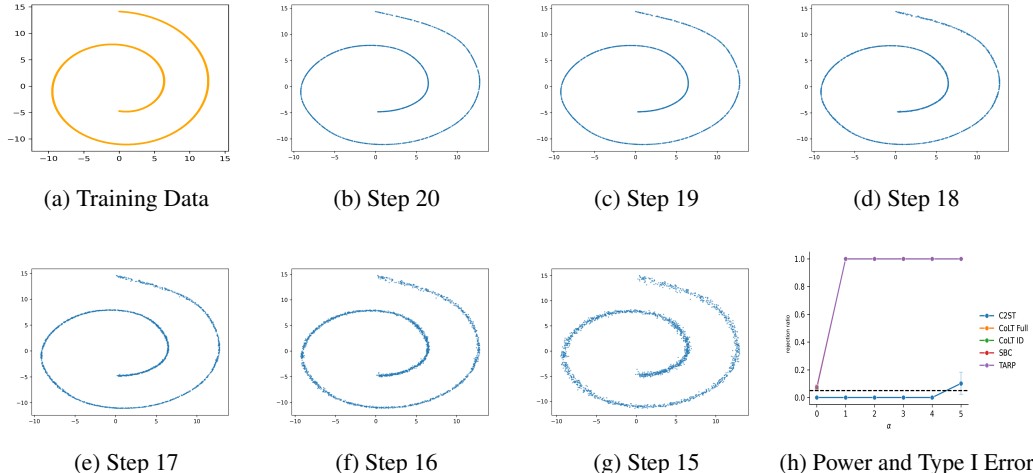

| (a) Training Data | (b) Step 20 | (c) Step 19 | (d) Step 18 |
| --- | --- | --- | --- |
| (e) Step 17 | (f) Step 16 | (g) Step 15 | (h) Power and Type I Error |

Figure 7: Visualization of the diffusion-based posterior sampling process. (a) shows the data manifold used for training. (b)–(g) show sampled distributions at various reverse diffusion steps (15 to 20), where step 20 is treated as the ground truth. (h) plots the statistical power and Type I error as a function of the step gap from the final posterior.

## E   Ablation Studies

In this section, we present additional ablation experiments to validate the design choices and examine the robustness of our proposed method. These studies explore the impact of different architectural and algorithmic components, providing a more comprehensive understanding of the method's performance across the design space.

**Model Capacity.**    As long as the localization and embedding networks possess sufficient representational capacity, we expect them to achieve comparable performance. As detailed in Section C, we employ 3-layer MLPs with 256 hidden units and observe consistent results across datasets of varying dimensionality and distributional complexity. This indicates that, given adequate model capacity, performance remains stable and robust.

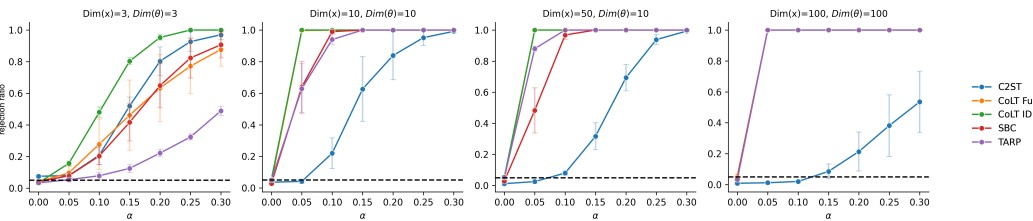

Figure 8: Mean Shift

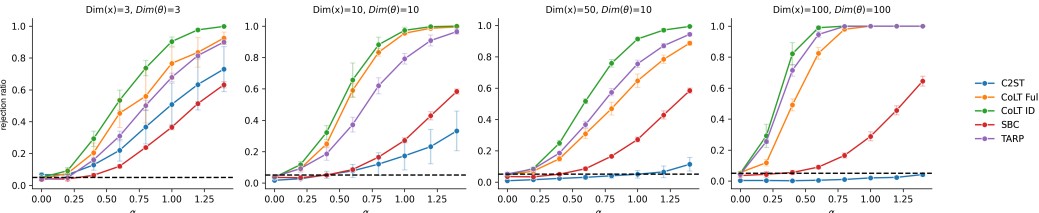

Figure 9: Covariance Scaling

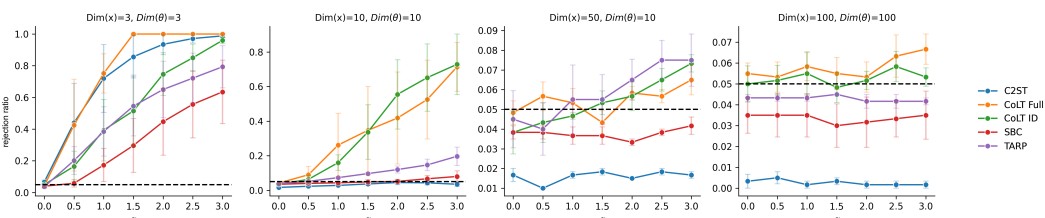

Figure 10: Anisotropic Covariance Perturbation

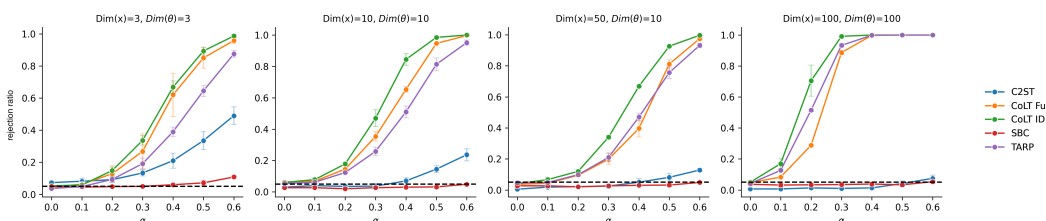

Figure 11: Kurtosis Adjustment via t-Distribution

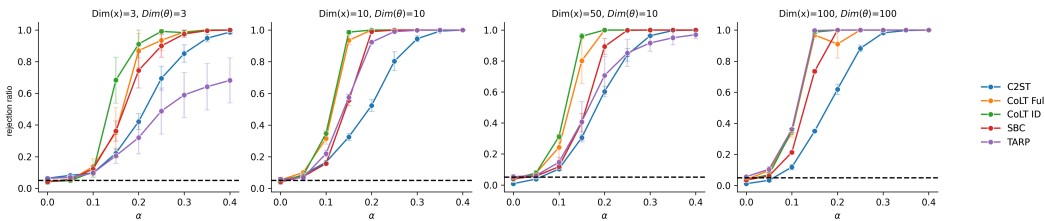

Figure 12: Additional Modes

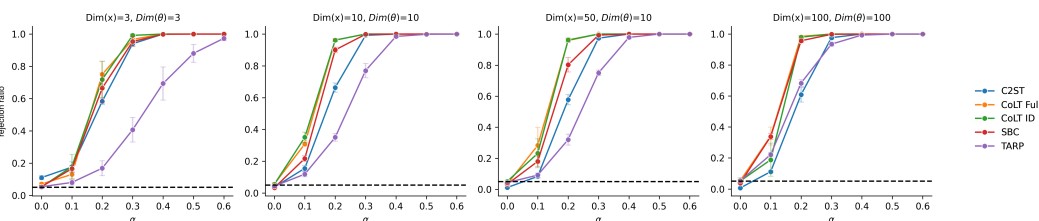

Figure 13: Mode Collapse

To further assess the effect of network depth, we conducted an ablation study under mean-shift perturbations with $\alpha = 0.2$ and $\alpha = 0.3$. The results, presented in Table 5, show that reducing the number of layers slightly weakens statistical power, while deeper networks yield marginal gains. Nevertheless, the overall performance remains within a similar range, suggesting diminishing returns beyond moderate depth. Although a full architecture search is beyond our current scope, we expect further improvements with more refined architectural design.

|  | Hidden Layer = 2 | Hidden Layer = 3 | Hidden Layer = 4 |
|---|---|---|---|
| $\alpha = 0.2$ (CoLT-Full) | 0.61 | 0.63 | 0.91 |
| $\alpha = 0.3$ (CoLT-Full) | 0.88 | 0.89 | 1.00 |

Table 5: Effect of MLP depth on the CoLT-Full performance under mean-shift perturbation with different $\alpha$. Increasing the number of hidden layers provides slight improvements, while maintaining overall consistency.

**Divergence Functions.** We conduct additional experiments using various divergence objectives and find that training remains stable across all variants, with the loss consistently decreasing over time. For instance, as shown in Table 6, in the kurtosis adjustment task under $t$-distribution perturbations with $\alpha = 0.2$ and $\alpha = 0.3$, the Sinkhorn divergence achieves the best performance. We attribute this to its smoother loss landscape, which facilitates optimization [16].

|  | Sinkhorn | MMD | Wasserstein | KS |
|---|---|---|---|---|
| $\alpha = 0.2$ | 0.13 | 0.08 | 0.09 | 0.10 |
| $\alpha = 0.3$ | 0.27 | 0.14 | 0.20 | 0.18 |

Table 6: Statistical power under different divergence objectives for the kurtosis adjustment task with $t$-distribution perturbations. The Sinkhorn divergence consistently outperforms others, likely due to its smoother and more optimization-friendly loss landscape.

