# OpenReview forum: "CoLT: The conditional localization test for assessing the accuracy of neural posterior estimates"
_NeurIPS.cc/2025/Conference — NeurIPS 2025 spotlight_

### Official Review · Reviewer_tRUQ · 2025-06-30

**Clarity:** 3
**Significance:** 3
**Originality:** 3
**Rating:** 5
**Confidence:** 4

**Summary:**

The paper aims to develop a method to determine if a posterior distribution corresponding to data is approximately the same as a neural posterior estimate. The CoLT algorithm introduced uses a KS style statistic that can practically approximated to determine similarity of distributions.

**Questions:**

What is the significance of Theorem 1? The discussion on line 171 seems like it would be quite useful by elimiting the need for an exhaustive search over $x$, but Algorithm 1 and 2 seem to using entirely Theorem 2 and 3.

**Ethical Concerns:**

["NO or VERY MINOR ethics concerns only"]

**Final Justification:**

All issues brought have been adressed in the rebuttle. I see no more major techincal problems with the paper.

**Limitations:**

yes

**Quality:**

2

**Strengths And Weaknesses:**

While I am not overly familar with this area, the method introduced seems to be very good. It seems quite novel and performs well on the experiements presented.


All problems I was able to find are technical in nature.

First some minor mistakes
- Line 144: $\phi$ is not a metric. I guess what is meant is that it induces a metric through the euclidean norm as is done on line 718. Although $d_\phi$ need actually be a metric if $\phi$ is not injective. Are embeddings here defined to be injective? If so, then it doesn't seem that theoretical results can be applied to the case where $\phi$ is a neural network.
 - Line 149: What is $D$? Presumably $\Theta\subseteq {\mathcal R}^{D}$.
 - Line 150: It is said that it is sufficient for $\phi$ to be Lipschitz continuous, but appeandix A only shows that it is sufficient that $\phi$ is bi-Lipschitz. I also can't find the condition that the balls of $\phi$ live inside of Euclidean ellipsoids of uniformly bounded eccentricity in appendix A. This condition feels false as stated and probably needs more explaination. What if $\Theta$ is an simply an ellipse? All balls will then trivially be inside an ellipse
 - Theorem 1, line 162: is it is unclear if measurability is WRT the Euclidean function on Theta, or on the metric $d_\phi$.
- Line 729: I can't see how this condition holds for all $\theta$ and for $R$ sufficently small. I would have guessed that this is an application on the out of line equation on line 720 where this condition holds for a.e. $\theta$ and for all $R$.
 - Line 730: If $\Theta$ has an isolated point or if $\phi$ is discontinuous, then I don't think that $m(B_\phi(\theta,R))>0$.
 - Line 776: What is the mapping $\alpha\mapsto R_{\theta_l(x)}(\alpha)$ a bijection into? I would guess that the structure of $\Theta$ and $\phi$ would change the the size of balls and hence the range of values $\alpha\mapsto R_{\theta_l(x)}(\alpha)$ could possibly take. From line 778 I'm guessing that it is supposed to be bijective into the space of all possible distances to $\theta_l(x)$.

There are some mistakes which seem fairly major to me
- line 728: I can't find the Lebesgue differentiation theorem for doubling measures in [16]. I'd assume a similar result to theorem 1.8 in "Lectures on Analysis on Metric Spaces" by Juha Heinonen, or the Lebesgue differentiation theorem on page 81 of https://homepages.uc.edu/~shanmun/book.pdf is being applied. In both these theorems the measure is defined over a metric measure space, but as stated above $d_\phi$ is not a metric without some type of injectivity condition on $\phi$. It will however be a pseudo-metric, but I doubt that the Lebesgue differentiation theorem holds for pseudo-metrics. I'd also assume there is an implicit assumption that $\phi$ is continuous so that the Lebesgue measure can be defined on $(\Theta, d_\phi$).
 - Line 733: I can't see why the inner integral be 0. What is assumed is essentially that the inner integral is be 0 on average. This doesn't seem to imply that the integral must be equal to 0 a.e. If $d_x$ where always positive (or always negative) this true, but it seems that $d_x$ will essentially always not be sign-definite.
 - Line 734: Invoking lemma 1 is not valid here as R is fixed and lemma 1 requires the integral equality to hold for all R. I think a sequence of $R_n\to 0$ needs to be taken.
 - Line 732: It doesn't seem that the fact that the nested integral is =0 for all measurable $\theta_l$ used. The proof seems to depend only on there existing only a single $\theta_l$ for which the integral is 0. This assumption seems far too weak for the result to hold. The proof doesn't seem to be following the intuitive idea given in line 165.
 - Line 776: I can't see why $\alpha\mapsto R(\alpha)$ is an injection if $\phi$ is not injective. It seems that possible that for many $\alpha$ $R(\alpha)$ would be infinite as the infimum defining $R$ would be the infimum over $\emptyset$. It seems like small changes in $r$ can lead to large changes in the size of balls in $d_\phi$. The doubling condition might save this but this is not obvious to me. A map like $\phi(x)=0$ seems to be a counter-example.

The paper is quite well written. There are also some minor formatting mistakes in the appendix
 - Line 721, 730: To be consistent with the notation below and above, I think B should be defined as $B_\phi$.
 - Line 733: I think the wrong equation is cited. There is no integral in equation 1.
 - Line 746: I think $\theta^\star$ should be $\theta^*$. Similar problem with the equation below line 748
 - Line 775: I think notation is being mixed. $U_q(\theta^* |x)$ should be $U_{q,x}(\theta^* )$. If not, then I don't know what $U_q(\theta^* |x)$ is

---

> ### Author Rebuttal · Authors · 2025-07-31
>
> We thank the reviewer for their insightful questions and apologise for any confusion. We agree that we need to be clearer and more precise about regularity conditions. The two main issues are the conditions of theorem 1, and the proof itself.  With the clarifications below (to be incorporated into the next version), we hope you agree that the main results from the paper are correct and precisely stated.
>
> **We first address your concerns about the metric $d_\phi$**. You are correct that $\phi$ is not a metric, but induces a metric $d_\phi$ under certain assumptions. We apologise for the lack of precision. You are also correct that $\phi$ must be injective. Otherwise $d_\phi$ can fail the requirement that $\theta_1 = \theta_2 \iff d_\phi(\theta_1, \theta_2) = 0$. We apologise for not including this condition explicitly, which created confusion. Clearly if $\phi$ is bi-Lipschitz (which we do discuss explicitly), then it is necessarily injective, and also induces a $d_\phi$ that satisfies the doubling condition.  But it is clearer to state the doubling-condition requirement on the metric without explicitly referring to $\phi$, and then to discuss separately how we might construct a metric satisfying that condition using a trainable neural network.
>
>  In revision, we will state the regularity conditions of Theorem 1 precisely and include a clear discussion of what trainable metrics meet the conditions. In our original approach a bi-Lipschitz $\phi$ is enough.  There are several neural network architectures that work; normalizing flows and RealNVP lead to invertible mappings. In practice, most networks with injective activations that do not have a "bottleneck" layer that reduces the dimension of the input typically satisfy injectivity in practice.  But other approaches also work.  For example, we can use a deep kernel-based characterization of distance as in [1]. For $\epsilon>0$, we might define the deep kernel-based distance as $d_k(\theta, \theta') = \sqrt{2(1 - k_\epsilon(\theta, \theta'))}$ where $k_\epsilon(\theta, \theta') = [(1-\epsilon)k(\phi(\theta), \phi(\theta')) + \epsilon]q(\theta, \theta')$, where $k$ and $q$ are Gaussian RBF kernels and $\phi$ is a Lipschitz continuous function of order $L$ (parameterized by neural networks). [1] show that $k_\epsilon$ is a characteristic reproducing kernel and hence the corresponding Hilbert space mapping defined by $k_\epsilon$ is injective. In our revised manuscript, we will include a proof that for any Lipschitz continuous function $\phi$, the metric $d_k$ satisfies a **local doubling condition**, meaning that for any fixed radius $R_0 > 0$, there exists a constant $C_{R_0} > 0$ such that for all $\theta \in \Theta$ and all radii $r \in (0, R_0]$, $\mu(B_k(\theta, 2r)) \le C_{R_0} \mu(B_k(\theta, r))$, where $\mu$ is the Lebesgue measure. If $\Theta$ is compact, the metric is uniformly doubling trivially. But even for locally doubling spaces, the Lebesgue differentiation theorem holds, as its proof hinges on a Vitali-type covering theorem, which we can show still applies under the weaker locally doubling condition.
>
>  [1] Liu, Feng, et al, "Learning deep kernels for non-parametric two-sample tests."
>
> > Concerns about the proof of theorem 1.
>
>  Thank you for bringing your concerns to our attention. Upon close inspection, our proof had a gap. Here we briefly explain how to rectify this and give a fully correct proof.  The condition
> $$
> \int_{\mathcal{X}} p(x)\Bigl[\int_{B(\theta_l(x),R)} d_x(\theta)d\theta\Bigr]dx = 0
> \quad\text{for all measurable }\theta_l\colon\mathcal{X}\to\Theta
> $$
>
> says that **no matter how we choose the ball center as a function of $x$**, the $p$–weighted average of the localized discrepancy vanishes. If we only allowed **constant** selectors, cancellations across $x$ could hide nonzero regions of the inner integral; the key of the proof is to adapt the center to each $x$ so that cancellations cannot occur. This mirrors many classic results in variational analysis and stochastic control, which confront similar problems of converting averaged statements into pointwise conclusions. There the use of measurable selectors is a standard device that aligns a choice variable with the pointwise extremum of an integrand.
>
> Due to space constraints we can only sketch the proof. We define $F_R(x,\theta)=\int_{B(\theta,R)}d_x(\theta')d\theta'$, and for any $R$, define the envelope $G_R(x)=\sup_{\theta}|F_R(x,\theta)|$. Now fix $R>0$. The goal is to rule out any cancellation across $x$ by
> aiming the center at the worst local discrepancy. To do this in a
> measurable way, we construct a countable dense set $\\{\vartheta_k\\}\subset\Theta$
> and, for each $n\in\mathbb{N}$, choose a measurable *$\varepsilon$-maximizer*
> $\theta_{R,n}(x)\in\\{\vartheta_k\\}$ satisyfing
> \begin{equation*}
> \bigl|F_R\bigl(x,\theta_{R,n}(x)\bigr)\bigr|
> \ge G_R(x)-\tfrac{1}{n}
> \qquad\text{for all }x.
> \end{equation*}
> Because the hypothesis holds for **every** selector, it holds for these
> $\theta_{R,n}$ and also for analogous selectors that nearly maximize $-F_R$.
> This yields the **inequality**
> $$
> \int_{\mathcal{X}} p(x)G_R(x)dx
> \le \frac{2}{n} \int_{\mathcal{X}} p(x)dx
> \qquad \text{for each } n.
> $$
>
> Now pass to the limit $n\to\infty$. The right-hand side tends to $0$,
> while the left-hand side is a nonnegative number (it is the integral of
> a nonnegative function). Hence
> \begin{equation*}
> \int_{\mathcal{X}} p(x)G_R(x)dx = 0.
> \end{equation*}
> Intuitively, this says: even after we **optimize** the center to expose
> the largest local discrepancy at each $x$, the weighted average of those
> maximal discrepancies is still zero. Since $p(x)>0$ almost everywhere and
> $G_R\ge 0$, this forces $G_R(x)=0$ for $p$-a.e. $x$; otherwise there would
> exist $\varepsilon>0$ and a set $A$ with $\int_A p>0$ on which $G_R\ge\varepsilon$,
> leading to $\int pG_R\ge \varepsilon\int_A p>0$, a contradiction.
> Thus, for such $x$ and this fixed $R$,
> \begin{equation*}
> \sup_{\theta\in\Theta}|F_R(x,\theta)|=0
> \quad\Longrightarrow\quad
> F_R(x,\theta)=0\ \ \text{for all }\theta\in\Theta.
> \end{equation*}
> Finally, pass from fixed radii to points by shrinking the balls and apply the Lebesgue Differentiation Theorem.
>
> > Line 149: What is $D$?
>
> You are correct that $D$ refers to the dimensionality of $\theta$. We will clarify this point.
>
> > Line 150: It is said that it is sufficient for $\phi$ to be Lipschitz continuous... What if is an simply an ellipse? All balls will then trivially be inside an ellipse
>
> You are correct that we need bi-Lipschitz. We will state this clearly in the revised version, and we thank the referee for the opportunity to clarify. Our mention of an ``ellipsoidal condition'' was a hand-wavy reference to the geometric requirement that the metric $d_\phi$ must not exhibit unbounded local distortion. This intuition is formalized precisely by the condition that the embedding $\phi$  be bi-Lipschitz, which is the essential requirement of our original approach. We were simply trying (and failing) to articulate the idea that the bi-Lipschitz property ensures that the infinitesimal balls under $d_\phi$ correspond to Euclidean ellipsoids with uniformly bounded eccentricity. We will remove references to the ellipsoidal conditional and make clear that one way to satisfy the theorem's conditions is with a bi-Lipschitz $\phi$.
>
> > Theorem 1, line 162: is it is unclear if measurability is WRT the Euclidean function on Theta, or on the metric $d_\phi$.
>
> We will clarify that it is measurable on $\mathcal{X}$. We assume $x \in \mathbb{R}^d$ and that $\theta_{l}(x)$ is a measurable function.
>
> > Line 729: I can't see how this condition holds for all $\theta$ and $R$ for sufficently small. I would have guessed that this is an application on the out of line equation on line 720 where this condition holds for a.e. $\theta$ and for all $R$ .
>
> You are correct and we will clarify this point in the revision.
>
> > Line 730: If $\phi$ is discontinuous...
>
> If $\phi$ is bi-Lipschitz, it is trivially continuous and the problem does not arise.
>
> > Line 776: What is the mapping $\alpha\to R_{\theta_l(x)}(\alpha)$ a bijection into? I would guess that the structure of $\Theta$ and $\phi$ would change the the size of balls and hence the range of values $\alpha\to R_{\theta_l(x)}(\alpha)$ could possibly take. From line 778 I'm guessing that it is supposed to be bijective into the space of all possible distances to $\theta_l(x)$.
>
> Yes, you are correct. It is a bijection into the space of all possible distances to $\theta_l(x)$. We will clarify this in the revised manuscript.
>
> > line 728: I can't find the Lebesgue differentiation theorem for doubling measures in [16]. I'd assume a similar result to theorem 1.8 in "Lectures on Analysis on Metric Spaces" by Juha Heinonen...
>
> You are right and we have corrected the reference. Per the discussion above, our bi-Lipschitz condition on $\phi$ is sufficient to guarantee that $d_\phi$ is a metric satisfying the doubling condition.
>
> > Line 776: I can't see why $\alpha\to R(\alpha)$ is an injection if $\phi$ is not injective. It seems that possible that for many $\alpha$, $R(\alpha) $would be infinite as the infimum defining would be the infimum over $\emptyset$. It seems like small changes in $r$ can lead to large changes in the size of balls in $d_\phi$. The doubling condition might save this but this is not obvious to me. A map like $\phi(x)=0$ seems to be a counter-example.
>
> Per the earlier discussion, you are right that injectivity is required, and guaranteed by the bi-Lipschitz condition.
>
> > Line 721,730,733, 746,775
>
> Thank you for pointing out these formatting mistakes, and that the equation should be the first equation in Theorem 1. You are right, and we will correct them in the revision.

---

> ### Author Response · Authors · 2025-08-06
> **updated proof of Theorem 1**
>
> For completeness, we want to include the full updated proof of Thm1, to fill out the sketch given in the initial rebuttal. The proof has two main steps. First, we fix $R>0$ and use a measurable selection argument to show that the inner integral is zero for almost every $x$. Second, we apply the Lebesgue Differentiation Theorem to show that the integrand itself must be zero almost everywhere.
>
> Step 1: Show ball integrals are zero for a.e.\ $x$.
> Let $d_x(\theta) = p(\theta\mid x) - q(\theta\mid x)$. For a fixed $R>0$, define the ball integral
> $$ F_R(x,\theta) := \int_{B(\theta,R)} d_x(\theta')\,d\theta'. $$
> The function $F_R(x,\theta)$ is a Carathéodory function: that is, it is measurable in $x$ for each fixed $\theta$ (by Fubini's theorem, as $d_x$ is continuous in $x$) and continuous in $\theta$ for each fixed $x$ (by the Dominated Convergence Theorem, as $d_x \in L^1_{\text{loc}}$ and $m(\partial B(\theta,R))=0$).
>
> Define the pointwise suprema:
>
> $$
> S_R^+(x) := \sup_{\theta\in\Theta} F_R(x,\theta)
> $$
>
> $$
> S_R^-(x)   := \sup_{\theta\in\Theta} \bigl(-F_R(x,\theta)\bigr)
> $$
>
> $$
> G_R(x)   := \sup_{\theta\in\Theta} |F_R(x,\theta)| = \max\{S_R^+(x), S_R^-(x)\}
> $$
>
> Since $\Theta \subset R^d$ is separable, let $D=[ \xi_k ]_{k \geq 1}$ be a fixed countable dense subset. Because $\theta \rightarrow F_R(x,\theta)$ is continuous and $D$ is dense, the suprema over $\Theta$ are equal to the suprema over $D$.
>
> Specifically, let $\Sigma_{R}^{+}(x) = \sup_{k \geq 1} F_R(x, \xi_k)$ and $\Sigma_{R}^{-}(x) = \sup_{k \geq 1} (-F_R(x,\xi_k))$.  These are measurable functions, since they are countable suprema of measurable functions, and we have $S_R^+(x) = \Sigma_R^+(x)$ and $S_R^-(x) = \Sigma_R^-(x)$ for all $x$.
>
> Now, we construct measurable $\epsilon$-maximizing selectors. Fix an integer $n$ and define
> $$
> k_n^+(x) = \min \left[ k \geq 1 : \ F_R (x, \xi_k ) \geq \Sigma_R^+(x)- 1/n \right]
> $$
>
> and
>
> $$
> k_n^-(x) = \min \left[  k \geq 1 : \ -F_R(x,\xi_k) \geq \Sigma_R^-(x)- 1/n \right]
> $$
>
> These minima are well-defined and finite. By the definition of the supremum, for any $\epsilon=1/n > 0$, the set of indices $k$ satisfying the condition is guaranteed to be non-empty. By the well-ordering principle, a non-empty subset of $\mathbb{N}$ has a minimum. The maps $x\mapsto k_n^\pm(x)$ are measurable, as the sets $\{x : k_n^+(x)=k\}$ are formed by measurable comparisons. Thus, the selectors $\theta_{R,n}^+(x) := \xi_{k_n^+(x)}$ and $\theta_{R,n}^-(x) := \xi_{k_n^-(x)}$ are well defined and measurable.
>
> By construction, these selectors satisfy:
> $$ F_R\bigl(x,\theta_{R,n}^+(x)\bigr) \geq \Sigma_R^+(x)-\tfrac1n \quad \text{and} \quad -F_R\bigl(x,\theta_{R,n}^-(x)\bigr) \ge \Sigma_R^-(x)-\tfrac1n \quad \text{for all }x. $$
> Applying the theorem's hypothesis with $\theta_l = \theta_{R,n}^+$ gives:
> $$ 0 = \int_{\mathcal{X}} p(x) F_R\bigl(x,\theta_{R,n}^+(x)\bigr)\,dx \geq \int_{\mathcal{X}} p(x)\left(\Sigma_R^+(x)-\tfrac1n\right)dx \implies \int_{\mathcal{X}} p(x)\Sigma_R^+(x)\,dx \le \frac{1}{n}\int_{\mathcal{X}} p(x)\,dx. $$
> Similarly, applying the hypothesis with $\theta_l = \theta_{R,n}^-$ gives:
> $$ 0 = \int_{\mathcal{X}} p(x) F_R\bigl(x,\theta_{R,n}^-(x)\bigr)\,dx \le \int_{\mathcal{X}} p(x)\left(-\Sigma_R^-(x)+\tfrac1n\right)dx \implies \int_{\mathcal{X}} p(x)\Sigma_R^-(x)\,dx \le \frac{1}{n}\int_{\mathcal{X}} p(x)\,dx. $$
> Since $G_R(x) \le S_R^+(x) + S_R^-(x) = \Sigma_R^+(x) + \Sigma_R^-(x)$, we have
> $$ \int_{\mathcal{X}} p(x)G_R(x)\,dx \le \int_{\mathcal{X}} p(x)\Sigma_R^+(x)\,dx + \int_{\mathcal{X}} p(x)\Sigma_R^-(x)\,dx \le \frac{2}{n}\int_{\mathcal{X}} p(x)\,dx = \frac{2}{n}. $$
> This must hold for all $n$, and so we must have $\int_{\mathcal{X}} p(x)G_R(x)\,dx = 0$. Since $p(x)>0$ and $G_R(x)\geq0$, we must have $G_R(x)=0$ for $p$-a.e.\ $x$. This means that for $p$-a.e.\ $x$, we have $F_R(x,\theta)=0$ for all $\theta \in \Theta$.
>
> Step 2: Apply the Lebesgue Differentiation Theorem.
> From Step 1, we know there is a set $X_0 \subset \mathcal{X}$ with $p(X_0)=1$ such that for any $x \in X_0$, $\int_{B(\theta,R)} d_x(\theta')\,d\theta' = 0$ for all $R>0$ and all $\theta \in \Theta$.
>
> The Lebesgue Differentiation Theorem for doubling spaces states that for any function $f \in L^1_{\text{loc}}(\Theta, m)$,
> $$ f(\theta) = \lim_{R\to 0} \frac{1}{m(B(\theta,R))} \int_{B(\theta,R)} f(\theta')\,d\theta' \quad \text{for } m\text{-a.e. } \theta. $$
> For any fixed $x \in X_0$, the function $\theta \mapsto d_x(\theta)$ is in $L^1_{\text{loc}}(\Theta)$ by hypothesis. Applying the theorem gives:
> $$ d_x(\theta) = \lim_{R\to 0} \frac{1}{m(B(\theta,R))} \int_{B(\theta,R)} d_x(\theta')\,d\theta' = \lim_{R\to 0} \frac{0}{m(B(\theta,R))} = 0 $$
> for $m$-a.e.\ $\theta \in \Theta$.
>
> Since this holds for every $x \in X_0$ where $\mu(\mathcal{X} \setminus X_0)=0$, we have $d_x(\theta)=0$ for $\mu \otimes m$-a.e.\ $(x,\theta)$. Thus $p(\theta \mid x) = q(\theta \mid x)$ for $\mu \otimes m$-a.e.\ $(x,\theta)$.

---

> > ### Author Response · Authors · 2025-08-06
> >
> > We hope our updated and complete proof addresses your concerns. Please don’t hesitate to let us know if any questions remain—we’d be happy to clarify further.

---

> > > ### Comment · Reviewer_tRUQ · 2025-08-07
> > >
> > > Thank you for the full proof. All of my problems have been adressed and I will update my rating.
> > >
> > > Just one new minor problem. In the new proof of Theorem 1, it is shown that for each $x\in\mathcal X_0$, there is a set $\Theta_x$ with $m(\Theta_x^c)=0$ such that for all $\theta \in \Theta_x$, $d_x(\theta)=0$. This will then imply that $d$ is non-zero on a subset of $M=(\mathcal X\setminus \mathcal X_0)\cup \{(x,\theta):x\in\mathcal X_0,\, \theta\in \Theta_x^c\}:=\mathcal X_0^c \cup N$. If this set is $(\mu\otimes m)$ is measurable, then $d$ will be equal to 0 $(\mu\otimes m)$-a.e. by a usual Fubini's argument:
> > > $$(\mu\otimes m)(M) = \mu(\mathcal X\setminus \mathcal X_0) + \int_{\mathcal X\times \Theta} 1_N(x,\theta) d(\mu\times m)(x,\theta) = \int_{\mathcal X} \int_{\Theta} 1_{\mathcal X_0}(X) 1_{\Theta_x}(\theta) dm(\theta) d\mu(x) =\int_{\mathcal X_0} m(\Theta_x^c) d\mu(x) = \int_{\mathcal X_0}0 d\mu(x) =0 $$
> > > But is $N$ measurable? It's sections definitely are, but measurability of sections doesn't imply measurable in the product. This is very nitpicky comment that I don't think necessarily needs to be adressed. It feels like it should be measurable (or at least its possible to get a similar set that is measurable) because $d$ is jointly measurable.

---

> ### Author Response · Authors · 2025-08-08
>
> We sincerely thank the reviewer for their positive reassessment of our work. We are glad that we could address all your concerns. Thank you for giving us your valuable feedback that helped improve the quality of our manuscript.
>
> ## Regarding the measurability of set $N$
>
> Thank you for raising the important technical question regarding the measurability of the set $N$. This is a subtle and crucial point for the validity of the Fubini's theorem argument. We are confident that the set is indeed measurable, and we provide the following detailed justification by constructing the set from measurable components.
>
> Our argument is founded on the joint measurability of the function $d(x, \theta) = p(\theta \mid x) - q(\theta \mid x)$. As the reviewer noted, this is a standard property, as conditional probability densities are required by definition to be measurable with respect to the product $\sigma$-algebra.
>
> The core of our proof is to show that the condition "$\theta$ is a non-Lebesgue point of $d_x$" corresponds to a measurable set in the product space. A point fails to be a Lebesgue point if the limit of its local averages either does not exist or does not equal the function's value. We formalize this construction as follows:
>
> 1.  **The Average Function.** For any radius $R>0$, let $f_R(x, \theta)$ be the average of $d(x, \cdot)$ over the ball $B(\theta, R)$:
>     $$
>     f_R(x, \theta) = \frac{1}{m(B(\theta, R))} \int_{B(\theta, R)} d(x, \theta') \, dm(\theta')
>     $$
>     Because $d(x, \theta)$ is jointly measurable, it is a standard result that $f_R(x, \theta)$ is also jointly measurable in $(x, \theta)$ for every fixed $R$.
>
> 2.  **The Limit Functions.** To handle the limit as $R \to 0$ while preserving measurability, we consider the limit over a countable sequence of radii (e.g., $R \in \mathbb{Q}^+$). We define the limit superior and limit inferior:
>     $$
>     L(x, \theta) = \limsup_{R \to 0, R \in \mathbb{Q}^+} f_R(x, \theta) \quad \text{and} \quad l(x, \theta) = \liminf_{R \to 0, R \in \mathbb{Q}^+} f_R(x, \theta)
>     $$
>     As the `limsup` and `liminf` of a countable collection of measurable functions, both $L(x, \theta)$ and $l(x, \theta)$ are jointly measurable functions.
>
> 3.  **The Set of All Non-Lebesgue Points.** The set of all pairs $(x, \theta)$ such that $\theta$ is a non-Lebesgue point of $d_x$ is the set $N'$. This set is composed of two cases: (a) where the limit does not exist, and (b) where the limit exists but is incorrect. It can be written as the union of two sets:
>     $$
>     N' = \{ (x, \theta) : l(x, \theta) \neq L(x, \theta) \} \cup \{(x, \theta) : d(x, \theta) \neq L(x, \theta)\}
>     $$
>     Since $d, l,$ and $L$ are all jointly measurable functions, the sets on the right-hand side are measurable with respect to the product $\sigma$-algebra $\mathcal{M} \otimes \mathcal{T}$. Therefore, their union $N'$ is also measurable.
>
> 4.  **Finalizing the Set $N$.** The set $N$ from our proof is the restriction of this general set $N'$ to the domain where $x \in X_0$. Formally, it is the intersection:
>     $$
>     N = N' \cap (X_0 \times \Theta)
>     $$
>     Since $X_0$ is a $\mu$-measurable set, the cylinder set $X_0 \times \Theta$ is measurable in the product space. As $N$ is the intersection of two measurable sets, it is itself measurable.
>
> This construction confirms that $N$ is a $(\mu \otimes m)$-measurable set, which fully justifies the application of Fubini's theorem in our proof. We hope this detailed explanation satisfactorily addresses the reviewer's concern and thank them again for their careful reading.

---

### Official Review · Reviewer_k1j4 · 2025-07-02

**Clarity:** 4
**Significance:** 3
**Originality:** 3
**Rating:** 5
**Confidence:** 4

**Summary:**

The authors propose a method for validating amortized variational posteriors; that is, those that define $q(\theta \mid x)$ for arbitrary $x$. The questions assessed is whether $p(\theta \mid x) = q(\theta \mid x)$ for all $x \in \mathcal{X}$. Although this would seem computationally intractable, the authors obtain tractability by i) assessing in expectation over $x$, ii)  using embeddings and localization points $\theta_l$ over which to assess the discrepancy between the two. The resulting algorithm involves training an embedding and localization point mapping on $x$ to *maximize* the discrepancy, and a KS test relative to a uniform distribution to make the final assessment.

**Questions:**

- One downside of the method is that small perturbations of $p$ are often OK; maybe even desirable (cf. line 103). For example, in the toy example, most practitioners would probably be thrilled if their variational posterior $q$ looked like panel (b) with target being panel (a).
- In fact, the NPE target (forward KL divergence), is known to yield overdispersed posteriors, and this has even been argued to be desirable for downstream applications such as importance sampling. This should be explained or at least touched on, maybe with reference to some of [1][2][3][4][5].
- It’s not particularly novel to consider an expectation over $p(x)$ instead of matching every single $x$, which is impossible (as noted by the authors). Indeed, NPE targets the expected forward KL divergence in the first place. Some type of reference to this fact should be given, preferably with citations (see below).
- Does $\Theta$ being compact suffice to satisfy the doubling condition?  If so, maybe this could be added to line 150.
- Theorem 1 seems pretty straightforward to me; I would recommend this be renamed a Proposition.
- The paper is light on citations; I would recommend citing some of the works below

[1] McNamara et al. Globally Convergent Variational Inference.

[2] Papamakarios and Murray. Fast $\epsilon$-free Inference of Simulation Models with Bayesian Conditional Density Estimation

[3] Le et al. Revisiting Reweighted Wake-Sleep for Models with Stochastic Control Flow

[4] Bornschein and Bengio. Reweighted Wake-Sleep

[5] Ambrogioni et al. Forward Amortized Variational Inference.

**Ethical Concerns:**

["NO or VERY MINOR ethics concerns only"]

**Final Justification:**

The authors adequately addressed all of my concerns, as well as those of other reviewers.

**Limitations:**

Yes

**Quality:**

3

**Strengths And Weaknesses:**

Strengths:
- The proposed method fills a gap in a literature by answering a pressing question in NPE that is as yet relatively unaddressed.
- The proposed method is motivated well theoretically; the uniform distribution of the ball probability rank and formulation of the IPM metric elegantly tie the result together.
- The exposition is clear and well-motivated intuitively.


Weaknesses:
- Any statistical “test” that involves as much machinery as training a neural network is a bit dissatisfying nominally in that it can not be easily applied  “out of the box”
- The motivation of the method is questionable in some areas: I would probably disagree with the assertion that  $p(\theta \mid x)$ and $q(\theta \mid x)$ need to match across all conditioning inputs $x \in \mathcal{X}$ (line 22). If the data space $\mathcal{X}$ is, for example, $\{0,1\}^{784}$, the space in which binarized MNIST digits would reside, we would only need these to match over a (practically) much smaller space. This is later implicitly accounted for by considering an expectation over $p(x)$ defined on $\mathcal{X}$, but I think the discussion around this point could be much improved.

---

> ### Author Rebuttal · Authors · 2025-07-30
>
> We thank the reviewer for the positive feedback and for recognizing the novelty, originality, and significance of our work. We address the reviewer’s concerns below.
>
> > W1. Any statistical “test” that involves as much machinery as training a neural network is a bit dissatisfying nominally in that it cannot be easily applied “out of the box”.
>
> 1. We use neural networks to parameterize the embedding and localization functions due to their flexibility in approximating a wide class of functions. **However, neural networks are not strictly necessary to our framework. One could replace them with alternative parametric families**, such as high-order polynomial function classes or other structured basis expansions. Since our divergence measure is based on the Sinkhorn distance, **it is, in principle, possible to estimate the embedding and localization functions analytically or via optimization techniques, without relying on neural network training**. While neural networks provide a convenient and expressive choice, our framework is general and compatible with other parameterizations that avoid training entirely.
>
> 2. Moreover, **while we use neural networks in our experiments, this does not necessarily imply a lack of "out-of-the-box" applicability**. Across a variety of experiments with different values of $x$- and $\theta$-dimensionality and varying posterior complexity, **we keep the neural network architecture fixed (3-layer MLPs with 256 hidden units per layer) and only make minor adjustments to hyperparameters** (see Section C.4). This consistency suggests that our method can be easily applied across a broad range of problems without the need for careful neural network design or hyperparameter tuning.
>
> 3. We appreciate the insight behind this question and agree that it is worth emphasizing. In our revision, we will include a discussion on how our framework could be instantiated using alternative parametric function classes that eliminate the need for training a neural network.
>
> > W2. The motivation of the method is questionable in some areas: I would probably disagree with the assertion that $p(\theta|x)$ and $q(\theta|x)$ need to match across all conditioning inputs $x \in \mathcal{X}$ (line 22). If the data space $\mathcal{X}$ is, for example, $\{0,1\}^{784}$, the space in which binarized MNIST digits would reside, we would only need these to match over a (practically) much smaller space. This is later implicitly accounted for by considering an expectation defined on $p(x)$, but I think the discussion around this point could be much improved.
>
> 1. Thank you for the insightful comment. We agree that this point deserves a more careful discussion, and we will include it in our revision.
>
> 2. The issue raised stems more from a **conceptual overgeneralization of the data space $\mathcal{X}$, which is not a limitation of our method but rather a consequence of an impractical choice of $\mathcal{X}$**. In simulation-based inference, $p(x)$ is defined through the generation process $p(\theta)p(x|\theta)$, meaning that the support of $p(x)$—and therefore the relevant domain of $\mathcal{X}$—is induced by the generator $p(x|\theta)$. For example, if $\theta$ indexes MNIST digits and $p(x|\theta)$ describes how images are generated, **then $\mathcal{X}$ corresponds to the manifold of MNIST-like images, not the entirety of $\\{0,1\\}^{784}$**. In this context, it is meaningful and appropriate to require $p(\theta|x) \approx q(\theta|x)$ for all $x$ in the support of $p(x)$.
>
> 3. Nevertheless, your point is important—it highlights that the practical choice of $\mathcal{X}$ should be considered carefully. We will revise the discussion to better clarify that our method targets alignment of posteriors over the support of the true data distribution, rather than over arbitrary or unrealistic regions of $\mathcal{X}$.
>
> > Q1. One downside of the method is that small perturbations of $q(\theta|x)$ are often OK; maybe even desirable (cf. line 103). For example, in the toy example, most practitioners would probably be thrilled if their variational posterior looked like panel (b) with the target being panel (a).
>
> 1. We would not consider this sensitivity to small perturbations a downside of our method. Rather, it reflects a deliberate trade-off: **if one aims to achieve high statistical power, the test must be sensitive to even small deviations from the true posterior**. This is **an inherent tension in hypothesis testing**—it is a double-edged sword, not a flaw of our method.
>
> 2. We fully understand the reviewer's concern that in practice, generative models often introduce acceptable approximations or slight deviations. **While our method may formally reject such approximate posteriors for a given significance level, the ACLD metric still provides a valid IPM-based distance between $q(\theta|x)$ and $p(\theta|x)$**. We further show this in Appendix C.3, where we use a diffusion model as a case study. **Although any non-zero perturbation level $\alpha > 0$ technically leads to rejection, the magnitude of the *KS test statistic* increases monotonically with $\alpha$ (see Figure 4)**. This behavior suggests that our method does more than issue binary accept/reject outcomes—it also offers a fine-grained measure of posterior closeness (by using KS test statistics).
>
> Therefore, even in applications where strict statistical matching is not the main goal, **our method can still serve as a useful diagnostic tool or distance metric to evaluate the quality of approximate inference procedures**.
>
> > Q2. In fact, the NPE target (forward KL divergence), is known to yield overdispersed posteriors, and this has even been argued to be desirable for downstream applications such as importance sampling. This should be explained or at least touched on, maybe with reference to some of [1][2][3][4][5].
>
> Thank you for pointing this out and for suggesting the relevant references—we will make sure to include and discuss them in our revision.
>
> 1. While overdispersed posteriors may be desirable in certain downstream tasks such as importance sampling, it remains important to assess how closely an approximate posterior $q(\theta \mid x)$ matches the true posterior $p(\theta \mid x)$. Our test serves as a principled test tool for such evaluations, **ensuring that the approximation is not excessively far from the ground truth.**
>
> 2. Even in settings where some degree of overdispersion is considered beneficial, **an approximate posterior that closely matches the true posterior is often ideal**. Passing our test indicates that $q(\theta \mid x)$ is sufficiently close to $p(\theta \mid x)$, which can lead to improved performance in downstream applications.
>
> > Q3. It’s not particularly novel to consider an expectation over instead of matching every single , which is impossible (as noted by the authors). Indeed, NPE targets the expected forward KL divergence in the first place. Some type of reference to this fact should be given, preferably with citations (see below).
>
> We agree that evaluating expectations over $x$ is not a novel idea, but it is also not the main contribution of our work. We appreciate the suggestion and will incorporate this clarification into the paper, along with citations to the relevant prior work.
>
> > Q4. Does ${\Theta}$ being compact suffice to satisfy the doubling condition? If so, maybe this could be added to line 150.
>
> Yes, the compactness of $\Theta$ is sufficient to ensure that the local doubling condition is satisfied. We will add this clarification in the revised version of the paper.
>
> > Q5.Theorem 1 seems pretty straightforward to me; I would recommend this be renamed a Proposition.
>
> Sure. Thanks for your suggestion, we would change it to 'Proposition' in the revision.
>
> > The paper is light on citations; I would recommend citing some of the works below
>
> Thank you for pointing these out. We will cite these literature in the revised version of the paper.

---

> > ### Comment · Reviewer_k1j4 · 2025-08-02
> > **Thanks for the detailed response**
> >
> > Thanks to the authors for their detailed response. They've adequately addressed my concerns, and I believe the changes will improve the paper. I'll closely monitor the discussion among the other reviewers throughout the rest of the discussion period.

---

### Official Review · Reviewer_ugqN · 2025-07-07

**Clarity:** 4
**Significance:** 3
**Originality:** 4
**Rating:** 5
**Confidence:** 3

**Summary:**

The paper presents *the conditional localisation test* (CoLT), a method for assessing the closeness of two conditional distributions. This is a common scenario in simulation-based inference, where one wishes to compare an approximate conditional posterior, $q(\theta | x)$, to the true conditional posterior, $p(\theta | x)$. The paper is based on the insight that if two distributions are unequal, they must differ in mass within a ball of some specific radius, and proposes to use a trained localisation function to find such a neighbourhood. The authors prove that finding this neighbourhood is possible (also empirically) through three theorems stating 1) that it is sufficient to compare the distributions' mass for an average over $x$ instead of for all $x$; 2) that the uniformity of a proposed ball probability rank statistic is necessary and sufficient to conclude if the distributions are identical; and 3) that finding the largest possible deviation from uniformity of this statistic provides an estimate of the distance between the distributions.

The authors provide algorithms for training and evaluating CoLT, which they apply to a benchmark suite of synthetic experiments, demonstrating that it works for a variety of conditional posteriors with varying complexity.

**Questions:**

1. Do you have some intuition (or practical experience) for how the capacity of the localisation and embedding networks should scale with the dimensionality and complexity of the posterior?
2. Is there a practical limit to the dimensionality of the problem that CoLT is useful for? As an extreme example, could CoLT be used to assess the performance of an implicit distribution over neural network parameters (i.e., millions or billions of parameters)?
3. You mention that $\phi$ must be Lipschitz, but I cannot find such a statement for $\theta_l(x)$ except that it should be smooth. Apologies if I missed it, but what are the requirements for the localisation function?
4. You mention that the localisation map can be used to expose where $q$ fails to match $p$, but I’m not sure how one would find these discrepancies in practice. Can you give an example?
5. Your theoretical results seem to suggest that CoLT provides a measure of the global discrepancy between distributions, but is this also the case in practice? When we optimise the localisation function, isn’t CoLT then restricted to assessing the local neighbourhood of that point?

**Ethical Concerns:**

["NO or VERY MINOR ethics concerns only"]

**Final Justification:**

The authors have addressed my questions and concerns, and given the paper's novel contribution and potential significance, I recommend acceptance.

**Limitations:**

Yes.

**Paper Formatting Concerns:**

None noticed.

**Quality:**

3

**Strengths And Weaknesses:**

**Strengths**
1. The paper presents a novel and original take on an important problem, which should be of wide interest to the NeurIPS community.
2. The authors are able to rigorously develop a practical algorithm for a seemingly intractable problem.
3. The paper is very well-written. I enjoyed reading it, in particular the authors' intuitive explanations of the theorems and the overall method.
4. The method seems likely to become a new benchmark for neural posterior estimation.

**Weaknesses**
1. While the experiments are interesting, they are also quite simple -- which the authors acknowledge too. I understand that it is difficult to define parametric, yet realistic posteriors for such benchmarks, but I am missing some more intuition for how to use the method in practice, for instance:
	1. How does the performance depend on the capacities of the localisation and embedding functions?
	2. Is there a practical limit to the dimensionality or complexity of the distributions where CoLT is effective?
	3. How is the stability of the training affected by different choices of the localisation and embedding functions and divergence measure?
2. It is unclear how computationally intensive the method is and how it compares to other methods.

---

> ### Author Rebuttal · Authors · 2025-07-30
>
> We thank the reviewer for the positive feedback and for recognizing the novelty, originality, and significance of our work. We address the reviewer’s concerns below.
>
> > W1.1-How does the performance depend on the capacities of the localisation and embedding functions?
>
> 1. As long as the localization and embedding networks have **sufficient capacity, we expect them to yield similar performance**. As discussed in **Lines 853–854**, we use 3-layer MLPs with 256 hidden units, and observe consistent performance across various data dimensionalities and distribution complexities. This suggests that, given adequate model capacity, performance remains robust.
>
> 2. We also conducted **an additional ablation study** using the mean shift perturbation with $\alpha=0.2$ and $0.3$ to examine the **impact of network depth**. Reducing the number of layers slightly decreases statistical power, while increasing layers slightly improves it. However, the overall performance remains in a similar range. While we did not perform a full architecture search, we believe further performance gains are possible with more carefully designed networks.
>
> |                         | **Hidden Layer = 2** | **Hidden Layer = 3** | **Hidden Layer = 4** |
> |-------------------------|----------------------|----------------------|----------------------|
> | $\alpha=0.2$ (CoLT-Full) | 0.61                 | 0.63                 | 0.91                 |
> | $\alpha=0.3$ (CoLT-Full) | 0.88                 | 0.89                 | 1.00                 |
>
>
> > W1.2-Is there a practical limit to the dimensionality or complexity of the distributions where CoLT is effective?
>
> 1.  Our experiments span a broad range of simulation-based inference settings, **from low-dimensional cases (e.g., $\dim(x)=3$, $\dim(\theta)=3$) to high-dimensional ones (e.g., $\dim(x)=100$, $\dim(\theta)=100$)**. We also evaluate CoLT on distributions of **varying complexity, including unimodal (e.g., Gaussian) and multimodal (e.g., Gaussian mixtures) cases**. As shown in Figure 3, Table 3, Table 4, and Figures 8-13, CoLT consistently performs well across these scenarios, and **we do not observe any inheren limitations tied to dimensionality or distributional complexity**.
>
> 2. In practice, **scaling to much higher-dimensional data (e.g., image domains) introduces challenges in optimization and training dynamics**, which may require **careful design choices such as data preprocessing, learning rate tuning, and appropriate architecture selection**. We agree that extending our method to these settings is a valuable direction for future work. Rather than viewing this as a limitation, we see it as a **domain-specific adaptation of our framework to new area**.
>
> > W1.3 – How is the stability of the training affected by different choices of the localisation and embedding functions and divergence measure?
>
> 1. Based on our experiments, **we do not observe any training instability resulting from different choices of localization or embedding functions**. These components are parameterized using a fixed neural network architecture (3-layer MLPs with 256 hidden units, as described in Lines 853–854), and **all experiments exhibit stable training behavior**. Furthermore, in our ablation study for W1.1, **varying the number of MLP layers does not lead to instability**. Our method is robust to architectural variations, and we perform only minimal hyperparameter tuning, as described in Section~C.4.
>
> 2. Similarly, **we do not observe training instability with different divergence measures. While the choice affects the final performance**, we conduct **additional experiments with different divergences** and find that training remains stable across all variants, with the training loss consistently decreasing. For example, in the kurtosis adjustment task using t-distribution perturbations with $\alpha = 0.2$ and $0.3$, the Sinkhorn divergence yields the best performance. As discussed in Lines 252–261, we attribute this to its smoother loss landscape, which is known to benefit optimization~[1].
>
>
> [1] Marco Cuturi. Sinkhorn distances: Lightspeed computation of optimal transport. Advances in
> 368 neural information processing systems, 26, 2013.
>
> |        | **Sinkhorn** | **MMD** | **Wasserstein** | **KS** |
> |:----------------:|:-------------:|:--------:|:----------------:|:-------:|
> | $\alpha=0.2$     | 0.13         | 0.08    | 0.09            | 0.10   |
> | $\alpha=0.3$     | 0.27         | 0.14    | 0.20            | 0.18   |
> Tab. Statistical power with different divergence loss.
>
> > W2.It is unclear how computationally intensive the method is and how it compares to other methods.
>
> We provide a **conceptual comparison of computational costs in terms of FLOPs** (floating point operations). Let $n$ be the number of samples from the true posterior, and for each $x$, let $m$ be the number of samples from the estimated posterior. Let $H$ denote the FLOPs required for a single forward pass of the neural network, and $E$ the number of training epochs. Ignoring dimensionality:
>
> - SBC and TARP require $\mathcal{O}(nm)$ FLOPs,
>
> - C2ST requires $\mathcal{O}(6nEH + 2nH)$ FLOPs,
>
> - CoLT requires $\mathcal{O}(6nmEH + 2nmH)$ FLOPs.
>
> While CoLT incurs higher training cost, the **neural networks we use are lightweight, and the actual wall-clock training time is short**—especially when compared to the computational cost of training the approximate posterior $q(\theta|x)$, which is typically parameterized by a generative model and requires significantly more computation resources.
>
> > Q1. ... how the capacity of the localisation and embedding networks should scale with the dimensionality and complexity of the posterior?
>
> 1. We evaluate CoLT across **a broad range of posterior dimensionalities (from 3 to 100) and complexities (unimodal to multimodal), and find that a 3-layer MLP with 256 hidden units is generally sufficient for reliable performance**. While increasing network capacity can improve power, we believe that in most statistical testing settings—typically involving moderate-dimensional posteriors—a 3-layer architecture is adequate. **For higher-dimensional domains such as image data, increased model capacity is necessary. In such cases, more expressive architectures like ResNet-18 may be appropriate replacements.**
>
> 2. Please also refer to our responses to W1.1 and W1.2.
>
> > Q2. Is there a practical limit to the dimensionality ... could CoLT be used to assess the performance of an implicit distribution over neural network parameters (i.e., millions or billions of parameters)?
>
> 1. When applying CoLT to more complex distributions—such as an implicit distribution over neural network parameters where both $\theta$ and $x$ may lie in extremely high-dimensional spaces—**the primary consideration becomes selecting a network architecture with sufficient capacity. In such cases, carefully designing the training dynamics and tuning the optimization hyperparameters become crucial for stable and effective training.**
>
> 2. While since our method ultimately relies on a ball-rank statistic (a one-dimensional summary), the localization and embedding **networks used in CoLT can be significantly lighter than the networks required to parameterize $q(\theta \mid x)$.**
>
> 3. Please also refer to our response to Q1, where we discuss the scalability and stability of our approach in more detail.
>
> > Q3. You mention that $\phi$ must be Lipschitz, but I cannot find such a statement for $\theta_l(x)$ except that it should be smooth....
>
> We do not need to make any assumptions about the regularity (e.g., smoothness) of $\theta_\ell(x)$. It is only required to be a Borel-measurable function over the space $\mathcal{X}$.
>
> > Q4. You mention that the localisation map can be used to expose where $q(\theta \mid x)$ fails to match $p(\theta \mid x)$, but I’m not sure how one would find these discrepancies in practice. Can you give an example?
>
> 1. Certainly. Suppose we are given a true posterior $p(\theta \mid x)$ and an approximate posterior $q(\theta \mid x)$, and we wish to identify where they differ most for a given $x$. After training, **we can feed $x$ into the localization network to obtain a point $\theta_{\ell}(x)$ in the parameter space**. This point corresponds to an anchor where the discrepancy between $p(\theta \mid x)$ and $q(\theta \mid x)$ is most pronounced.
>
> 2. Once this anchor is obtained, **we define a neighborhood (e.g., a ball) around it in the embedding space induced by the learned distance embedding $\phi$**. Given an input $x$, the localization and embedding networks together allow us to identify a local region where $p(\theta \mid x)$ and $q(\theta \mid x)$ differ the most, enabling more targeted and interpretable comparison.
>
> > Q5.Your theoretical results seem to suggest that CoLT provides a measure of the global discrepancy between distributions, but is this also the case in practice? When we optimise the localisation function, isn’t CoLT then restricted to assessing the local neighbourhood of that point?
>
> This gets to the heart of how CoLT operates. **While the optimization does indeed focus on a specific location, the resulting value is a valid measure of discrepancy**. To clarify, it's helpful to contrast two types of discepancy measures: ones like KL divergence, which compute a discrepancy based on an expectation (an average) over the entire sample space; and IPMs based on a supremum, such as the Kolmogorov-Smirnov (KS) distance. **Although the KS calculation hinges on finding a specific worst-case location, the resulting distance D is a well-established global metric for the discrepancy between distributions P and Q**.  CoLT operates on this same principle. The optimization of the localization function is precisely the search for this point of maximal local discrepancy. By finding the location and scale where the two measures differ most, CoLT defines a global discrepancy score based on this worst-case local separation.

---

> > ### Comment · Reviewer_ugqN · 2025-08-06
> >
> > Dear authors.
> >
> > Thank you for your comprehensive rebuttal, which addressed my questions and concerns. Your ablation study on the impact of the network depth is really interesting, but to me, the network capacity does seem to have quite some influence on the statistical power (e.g., when comparing two hidden layers to four). I don't think this is necessarily a weakness of CoLT, but I do think it warrants a brief discussion in the paper (or appendix). I hope you will incorporate that, as well as your answers to my comments and questions, into the final manuscript. I will keep my score for now and follow the discussion with reviewer tRUQ.

---

### Note · Authors · 2025-08-11

We think that the discussion during the rebuttal reflected a consensus that all issues raised by the reviewers were resolved, and we thank everyone for their suggestions to improve the paper!

---

### Decision · Program_Chairs · 2025-09-17

**Decision:**

Accept (spotlight)

**Comment:**

The paper proposes a method to detect discrepancies between approximate neural posteriors and the true ones. The method, conditional localization test (CoLT), trains localization functions to find neighborhoods of such discrepancy, which is supported by theoretical contributions regarding the possibility of this procedure. The latter is also empirically corroborated. These constitute the main strengths of the paper, a rigoros method with theoretical and empirical support for an important problem that has the potential to impact many other works. The present main weakness of the paper may be that it can be challenging to apply to a broader set of problems beyond the simple experiments shown in the paper. This seems to primarily be due to the fact that it can be very difficult to choose $\mathcal{X}$ appropriately in practical problems.

Overall, the strengths of the paper clearly outshine all the weaknesses. All reviewers rate the paper very positively, with a  few major concerns having being resolved in the rebuttal. One such concern was the practicality of the method to e.g. images, and the entailed discussion on dimensionality of the input. The authors have agreed to resolve this through further discussion in the paper and the AC recommends to include it in the camera ready. The other main concern was related to several assumptions being insufficiently discussed in the paper writing, which has led to reviewer tRUQ initially questioning the validity of some of the derivations. Through engaged discussion, these concerns have been resolved and additional explanations will be provided in the paper. The AC agrees with all reviewers and believes this paper is a valuable contribution to a meaningful problem that many might care about in the community.

The AC recommends to accept the paper with a spotlight.